# Dissecting transition cells from single-cell transcriptome data through multiscale stochastic dynamics

Peijie Zhou[1,2], Shuxiong Wang [2], Tiejun Li [1✉] & Qing Nie[2,3✉]

Advances in single-cell technologies allow scrutinizing of heterogeneous cell states, however, detecting cell-state transitions from snap-shot single-cell transcriptome data remains challenging. To investigate cells with transient properties or mixed identities, we present MuTrans, a method based on multiscale reduction technique to identify the underlying stochastic dynamics that prescribes cell-fate transitions. By iteratively unifying transition dynamics across multiple scales, MuTrans constructs the cell-fate dynamical manifold that depicts progression of cell-state transitions, and distinguishes stable and transition cells. In addition, MuTrans quantifies the likelihood of all possible transition trajectories between cell states using coarse-grained transition path theory. Downstream analysis identifies distinct genes that mark the transient states or drive the transitions. The method is consistent with the well-established Langevin equation and transition rate theory. Applying MuTrans to datasets collected from five different single-cell experimental platforms, we show its capability and scalability to robustly unravel complex cell fate dynamics induced by transition cells in systems such as tumor EMT, iPSC differentiation and blood cell differentiation. Overall, our method bridges data-driven and model-based approaches on cell-fate transitions at single-cell resolution.

[1] LMAM and School of Mathematical Sciences, Peking University, Beijing, China. [2] Department of Mathematics, University of California, Irvine, Irvine, CA, USA. [3] Department of Cell and Developmental Biology, University of California, Irvine, CA, USA. ✉email: tieli@pku.edu.cn; qnie@uci.edu

Advances in single-cell transcriptome techniques allow us to inspect cell states and cell-state transitions at fine resolution[1], and the notion of *transition cells* (aka. hybrid state, or intermediate state cells) starts to draw increasing attention[2–4]. Transition cells are characterized by their transient dynamics during cell-fate switch[3], or their mixed identities from multiple cell states[5], different from the well-defined stable cell states[6,7] that usually express marker genes with distinct biological functions. Transition cells are conceived vital in many important biological processes, such as tissue development, blood cell generation, cancer metastasis, or drug resistance[8].

Despite the rapid algorithmic progress in single-cell data analysis[9], it remains challenging to probe transition cells accurately and robustly from single-cell transcriptome datasets. Often, the transition cells are rare and dynamic, and herein difficult to be captured by static dimension-reduction methods[10]. High-accuracy clustering methods (e.g., SC3[11] and SIMLR[12]) tend to

enforce distinct cell states, placing transient cells into different clusters, therefore only applicable to the cases of sharp cell-state transition (Fig. 1a). While popular pseudo-time ordering methods[13], such as DPT[7], Slingshot[14] and Monocle[15], presumes either discrete (Fig. 1a) or continuous cell-state transition (Fig. 1a), quantitative discrimination between stable and transition cells is lacking[7]. Recently, soft-clustering techniques provides a way to estimate the level of mixture of multiple cell states[16], however, the linear or static models embedded in such approach make it difficult to capture dynamical properties of cells.

Dynamic modeling provides a natural way to characterize transition cells[3], allowing multiscale description of cell-fate transition (Fig. 1a and Supplementary Fig. 1). Such models analogize cells undergoing transition to particles confined in multiple potential wells with randomness[17,18], for which the transient states correspond to saddle points and the stable cell states correspond to attractors[19–21] of the underlying dynamical system

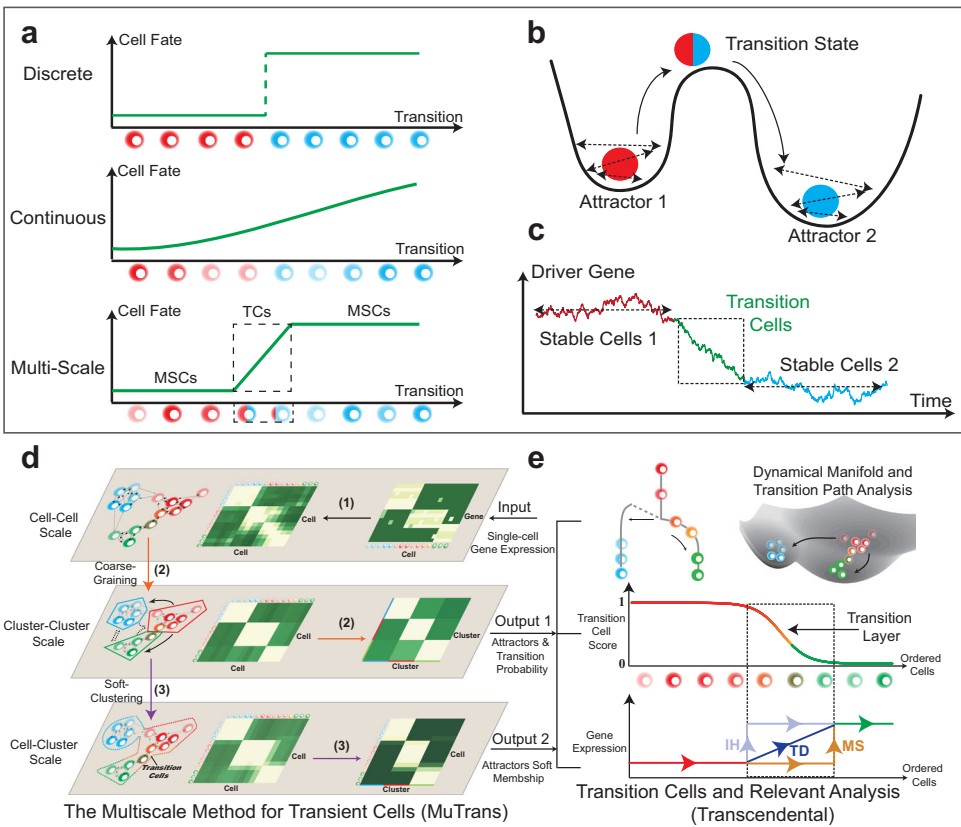

**Fig. 1 Brief introduction to MuTrans. a–c** Theoretical foundation of MuTrans - the multi-scale stochastic dynamics approach to model cell-fate transitions. **a** Three possible perspectives to describe cell-fate transition, as either entirely discrete or continuous process, or as the multi-scale switch process between attractors mediated by transition cells. The first two perspectives correspond to clustering or pseudotime ordering commonly adopted in single-cell analysis. **b** Biophysical foundation of the multi-scale perspective to treat cell-fate transition as over-damped Langevin dynamics in the multi-stable potential wells. The stable states correspond to the attractor basins while the transition states are modelled by the saddle points of underlying dynamical system. **c** A typical gene expression trajectory of multi-scale dynamics. The expression of driver genes fluctuates within the stable cells, while witnesses the continuous yet temporary change within transition cells, forming a transition layer in trajectory. **d, e** The procedure and downstream analysis of MuTrans. **d** The procedure of iterative multi-scale learning. The input is the pre-processed single-cell gene expression matrix. The three major steps (indicated by the number on arrow) for iterative learning of the stochastic dynamics across three different scales: (1) learning the cell-cell scale random walk transition probability matrix (rwTPM) from expression data, (2) learning the cluster-cluster scale rwTPM by coarse-graining the cell-cell scale rwTPM, and (3) learning the cell-cluster scale rwTPM by soft-clustering the cluster-cluster scale rwTPM. The output of iterative multi-scale learning includes the cell attractor basins and their mutual transition probabilities, as well as the membership matrix indicating relative cell positions in different attractors. **e** Downstream analysis (Transcendental Procedure). Given the output of iterative multi-scale learning, MuTrans constructs the cell lineage, dynamical manifold and transition paths manifesting the underlying transition dynamics of cell-fate. For each state-transition process, MuTrans explicitly distinguishes between stable and transition cells via transition cell score (TCS). The transition cells are marked with dashed squares. Based on the TCS ordering of cells, MuTrans identifies three types of genes (MS, IH and TD) during the transition whose expression dynamics differ in stable and transition cells. MS: meta-stable genes. IH: intermediate-hybrid genes. TD: transition-driver genes.

(Fig. 1b). In such description, the stochastic gene dynamics at individual cell scale can induce cell-state switch at macroscopic cell cluster or phenotype scale, and the transition cells form bridges between different attractors (Fig. 1c). Despite widely use of dynamical systems concepts to illustrate cell-fate decision[4], direct inference via dynamical models for transitions from single-cell transcriptome data is lacking.

Here we employ noise-perturbed dynamical systems[22] with a multiscale approach on cell-fate conversion[23] to analyze single-cell transcriptome data. By characterizing stable cells in attractor basins and placing the transition cells along transition paths connecting attractors through saddle points, our multiscale method for transient cells (MuTrans) prescribes a stochastic dynamical system for a given dataset (Fig. 1b). Using the single-cell expression matrix as input, through iteratively constructing and integrating cellular random walks across three scales (Fig. 1d and Supplementary Fig. 2), MuTrans finds most probable transition paths for cell transitions in a reconstructed cell-fate dynamical manifold (Fig. 1e, Methods). Such manifold, similar to the classical Waddington landscape[24] often used to highlight transitions, provides an intuitive visualization of cell dynamics compared to commonly adopted low-dimension geometrical manifold. In the dynamical manifold, the barrier height naturally quantifies the likelihood of cell-fate switch, and the Transition Cell Score (TCS) and transition entropy allows us to distinguish between attractors and transition cells (Fig. 1e, Methods). We then illustrate the complex cell transition trajectories on dynamical manifold using the dominant transition paths obtained for the coarse-grained dynamics. With such quantification, we are able to identify critical genes that are transition drivers (TD genes), mark the intermediate/hybrid states (IH genes) or meta-stable cells (MS genes) (Fig. 1e and Supplementary Fig. 3). To speed up calculations for datasets consisting of large number of cells[25,26], MuTrans provides an additional (and optional) aggregation module in pre-processing. This module aggregates cells into many small groups that share similar dynamical properties, thus MuTrans can take the transition probabilities among these coarse-grained cells as the input, instead of the random walk on original cells, in order to reduce the computational cost (Method and Supplementary Note 2).

We demonstrate the effectiveness and robustness of MuTrans in multiple single-cell transcriptome datasets, including simulation datasets and sequencing data generated by five different experimental platforms. Comparisons with existing single-cell lineage inference tools demonstrate the capability and scalability of MuTrans in probing complex, sometimes subtle, cell-fate transition dynamics. We also perform mathematical analysis to show consistency of MuTrans with the over-damped Langevin dynamics[27] - a popular model for state transitions in physical or biochemical systems[22].

## Results

**Overview of MuTrans workflow and theoretical foundations.** MuTrans depicts cells and their transitions in each single-cell transcriptome dataset as a multiscale dynamical system (Fig. 1a–c). The dynamics of cell fates can be described by the stochastic differential equations (SDEs) as

$$d\mathbf{X}_t = \mathbf{f}(\mathbf{X}_t)dt + \boldsymbol{\sigma}(\mathbf{X}_t)d\mathbf{W}_t, \quad (1)$$

where $\mathbf{X}_t \in \mathbb{R}^p$ denotes the cell's gene expression state at time $t$, $\mathbf{f}(x)$ denotes the nonlinear gene regulations, $\boldsymbol{\sigma}(x)$ denotes the noise strength due to both biochemical reactions and environmental fluctuations, and $\mathbf{W}_t$ is the standard Brownian motion representing the noise. Usually, $\mathbf{f}(x)$ may have multiple zeros, corresponding to the multi-stable attractors of the dynamical

system. At long time scale in coarse-grained state space, the Eq. (1) can be reduced to capture the transitions among different attractors[28].

To ensure the description is well-posed for single-cell transcriptome data, regularizations or additional prior knowledge (e.g., cell growth rate) needs to be enforced or provided. Similar to previous studies[29], here we make two important assumptions: (a) The multi-stable drift term $\mathbf{f}(x)$ can be well-approximated by the gradient of a potential field with multiple wells, and (b) the single-cell data is sampled from nearly stationary distribution (or a system is fully ergodic without rapidly growing populations). This indicates that the data is sampled from a stationary system, a reasonable assumption if no prior knowledge is provided[29]. From the decomposition analysis of differential equations[30], the potential-field assumption (a) is valid when the non-gradient term of drift $\mathbf{f}(x)$ is small in the large regions of state space, which holds in many biological systems with multi-stability[31]. Computationally, instead of fitting or solving the high-dimensional Eq. (1) directly, here we recover the dynamical structure of its solution using a multi-scale data-driven approach, as described below.

Taking the input as pre-processed single-cell gene expression matrix, MuTrans first learns the cellular random walk transition probability matrix (rwTPM) on the cell-cell scale through a Gaussian-like kernel (Fig. 1d and Methods), which yields the continuous limit as over-damped Langevin equation (Methods and Supplementary Note 1). Enforced by Gaussian-like kernel, the constructed rwTPM is in detailed-balance, consistent with the assumption (a). Next, the method performs coarse-graining on the cell–cell scale rwTPM to learn the dynamics on the cluster-cluster scale, and acquires attractor basins and their mutual conversion probabilities simultaneously (Fig. 1d and Methods). Theoretically, this step is asymptotically consistent with the Kramers' law of reaction rate for over-damped Langevin equations if assumption (b) holds (Methods and Supplementary Note 1). Finally, we specify the relative position of each cell in the attractor basins with the cell-cluster resolution view of Langevin dynamics, which is constructed via optimizing a cell-cluster membership matrix (Fig. 1d and Methods).

To robustly depict the lineage relationships, we use the transition path theory to quantify the likelihood of all possible transition trajectories between cell states, based on the coarse-grained transition probabilities (Fig. 1e, Methods and Supplementary Note 2).

Combining the optimized cell-cluster membership matrix, MuTrans fits a dynamical manifold using a mixture distribution to make stable cells reside in the attractor basins while assign transition cells along the transition paths connecting different basins (Fig. 1e and Methods), which is based on the Gaussian mixture approximation toward the steady-state distribution of the Fokker-Planck equation associated with the over-damped Langevin dynamics (Methods and Supplementary Note 2).

For each cell-state transition, we can calculate a transition cell score (TCS) ranging between one and zero to quantitatively distinguish attractors and transition cells (Fig. 1e and Methods). Finally, we systematically classify three types of genes (MS, IH and TD) during the transition whose expression dynamics differ between stable and transition cells (Fig. 1e and Methods). Specifically, the TD genes varies accordingly with the TCS within transition cells, and the IH genes co-express in both stable and transition cells, while MS genes express uniquely near the attractors.

To deal with the large-scale datasets, in addition to common strategies such as sub-sampling cells, we provide an option to speed up calculation by introducing a pre-processing aggregation module DECLARE (dynamics-preserving cells aggregation). This

module assigns the original individual cells into many (e.g., hundreds or thousands) microscopic stable states and computes the transition probabilities among them, and thus it can be used as an input to MuTrans instead of the cell-cell rwTPM (Methods and Supplementary Note 2). Both theoretical and numerical analysis suggest that, compared to the common strategy of averaging of gene expression profiles of a small group of cells, DECLARE better preserves the structure of dynamical landscape with a good approximation to the transition probabilities calculated without using DECLARE (Methods and Supplementary Note 2).

**Evaluation of MuTrans using simulation datasets**. To test accuracy and robustness of MuTrans, we evaluated its performance on simulation datasets generated from known dynamical systems. First we simulated the stochastic state-transition process using a bifurcation model in the regime of intermediate noise level[32]. The gene expression of each cell was simulated with over-damped Langevin equation driven by an extrinsic signal and

noise (Supplementary Note 3). In certain parameter range, the model consists of two stable states and one saddle state (Fig. 2a). Noise in gene expression induced the switch prior to the bifurcation point, resulting in a thin layer of transition cells (Fig. 2a). Applying MuTrans to the known transition cells and stable cells in the model, we found the computed transition cell score (TCS) captured the underlying saddle-node bifurcation structure (Fig. 2a). For cells fluctuating around the two stable branches, the TCS approaches one or zero respectively, indicating the meta-stability of cell states. The transition cells that pass the saddle point region in the trajectory yields a continuum of TCS between zero and one, with scores consistent with the relative positions of cells along the trajectory (Fig. 2a).

In addition to the uni-directional transition simulation dataset, we next consider back-and-forth stochastic state-switching, a common scenario in multi-stable systems. We constructed a triple-well potential field and simulated the dynamics with over-damped Langevin equations (Supplementary Note 3). Three saddle points lie between the attractor basins in its potential field,

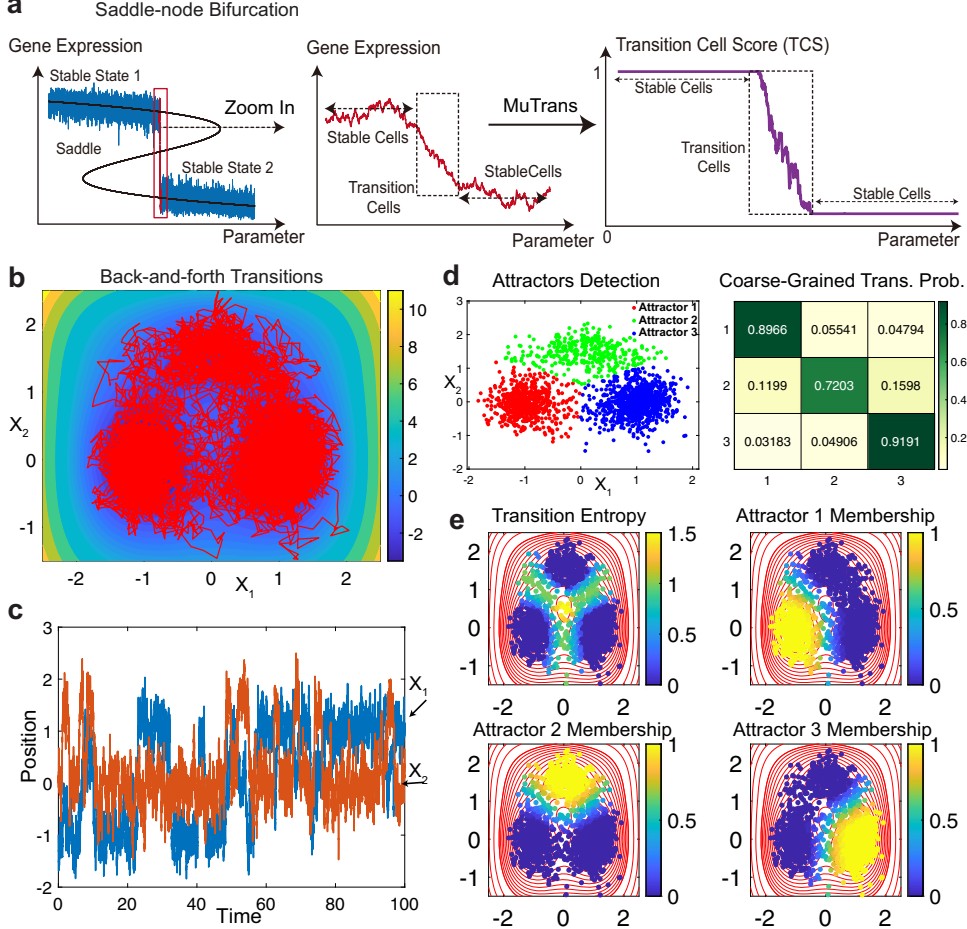

**Fig. 2 Evaluation of MuTrans in dynamical system simulation datasets. a** MuTrans distinguishes the stable and transition cells simulated using a stochastic saddle-bifurcation model. (Blue lines) The simulated trajectories of model. (Black Lines) Bifurcation plot of the underlying dynamical system. (Red Lines) The trajectory points corresponding to the transition cells that are switching between two states as the input to MuTrans ($N$=2,000). (Purple Lines) The transition cell score (TCS) values for transition cells calculated by MuTrans. The stable cells have TCS of value 0 or 1, while the TCS of transition cells decrease from 1 to 0 during transition. **b, c** The simulation dataset of over-damped Langevin dynamics in two-dimensional, triple potential-well system. **b** Simulated trajectories (red lines) and potential field (values indicated by color bars) in two-dimensional phase space. **c** Time series of the simulated trajectories, where the abrupt changes of values indicate state-transitions among attractor basins. **d, e** MuTrans reveals the transitions in triple-well system solely from sampled, snap-shot datasets ($N$ = 2,001). **d** The method detects three attractor basins and coarse-grained transition probability matrix among them. Cells are colored by attractors. **e** The transition cells near saddle points have larger MuTrans transition entropy than the cells near fixed points. The calculated membership functions quantify the relative cell positions in each attractor. Entropy and membership values are indicated by color bars.

with another maximum point (order-2 saddle) at the origin (Fig. 2b). Time series indicate that the state-hopping among three attractor basins can be frequent when large enough noise amplitudes are used (Fig. 2b, c). Using the simulation trajectories as snap-shot data points for inputs, MuTrans correctly infers three attractor basins from the dataset (Fig. 2d and Supplementary Fig. 4). The coarse-grained transition probabilities (Fig. 2d) suggest that the cells most likely remain in their original attractors other than transitioning into other attractors.

Consistent with previous studies of similar systems[33], as noise amplitude increases, the transitions between different attractors become more frequent as indicated by the larger coarse-grained probabilities. The direct transitions between attractors 1 and 3 are dominant compared with the state-switch mediated by attractor 2 (Supplementary Fig. 4). The calculated transition entropy and attractor membership functions accurately highlight the transition cells moving across various saddle points (Fig. 2e). Interestingly, the cells near the global maximum point have larger transition entropies than those near the first-order saddle points, indicating more mixed or hybrid identities.

**Revealing the cell-state transitions during EMT of squamous cell carcinoma**. We then applied MuTrans to a single-cell RNA sequencing dataset[34] of Squamous Cell Carcinoma (SCC) epithelial-to-mesenchymal transition (EMT) generated by Smart-Seq2 platform (Fig. 3 and Supplementary Fig. 5). Five attractors are detected by MuTrans (see Supplementary Fig. 5b for the corresponding EPI analysis), including one epithelial state (E), two mesenchymal states (M1 and M2) and two intermediate cell states (ICS). The cell states are annotated by comparing marker genes expression with those in the original study (Fig. 3a–d and Supplementary Fig. 5). Streams of transition cells moving between various attractor basins are observed in the constructed dynamical manifold (Fig. 3e).

The transition path analysis shows the major portion of the transition flux (more than 50%) from E state to M states goes through one of the ICS (Fig. 3f, g), indicating the significant role of ICS to mediate state-transitions in EMT[35,36]. Interestingly, there are also transitions within the two mesenchymal attractors, an observation consistent with the concept of quasi-mesenchymal states reported in the original study, suggesting that the M attractors here may also serve as intermediate nodes in transitions.

The transition gene analysis along the path E-ICS2-M2 characterizes the transition cells in their gene expression dynamics (Fig. 3h, i). Compared with MS genes that are highly expressed in stable cells, the IH genes may express in both transition cells and stable cells. The expressions of TD genes vary gradually within the transition cells (Fig. 3h, i).

**Scrutinizing bifurcation dynamics during iPSC induction**. We next used MuTrans to investigate cell fate bifurcations (Fig. 4a) in a single-cell dataset for induced pluripotent stem cells (iPSCs) toward cardiomyocytes[37]. In the learned cellular random walk across different scales, the rwTPM on cell-cluster scale recovers finer resolution of rwTPM on the cell-cell scale than the cluster-cluster scale (Fig. 4b). MuTrans identified nine attractor basins under this resolution (Fig. 4c and Supplementary Fig. 6), and the constructed tree (Supplementary Fig. 6) reveals a lineage with bifurcation into mesodermal (M) or endodermal (En) cell fates. Two attractor basins, locating before the bifurcation of primitive streak (PS) into differentiated mesodermal (M) or endodermal (En) cell fates, are denoted as Pre-M and Pre-En states (Fig. 4d and Supplementary Fig. 7). On the inferred dynamical manifold (Fig. 4e–g), the cells make transitions between two states,

suggesting possible dynamic conversion between the two types of precursor cells that seem to be very plastic. In comparison, the transition between mature En and M states are rare, indicating the stability of En and M cells. Along the differentiation trajectory from PS to Pre-M, the coarse-grained transition probability, quantified by the heights of barrier, shows a stronger transition capability from PS to Pre-M than from Pre-M to PS (Fig. 4c). In addition, the transition from Pre-M to M was found to be sharper than the one from PS to Pre-M. The transitions from PS to Pre-En and from Pre-En to En exhibit similar behavior. This analysis suggests that the initial cell-fate bifurcation at PS state (mostly on day 2–2.5, Fig. S6) is not terminal. This is consistent with the transition path analysis (Fig. 4e), showing that prior to the final commitment into M fate, some cells in PS take a detour by passing through the pre-En attractor basins first. The trend of transition entropy defined by MuTrans is found to be consistent with the critical transition index defined in original publication[37] for bifurcations. Indeed, the MuTrans transition entropy of cells first increases toward the bifurcation point from day 1 to 2.5, and then decreases as the final cell-fates are committed and established at day 3 (Fig. 4f, S6).

Downstream analysis on gene expression profiles indicates three transition stages from Pre-M to M (Fig. 4h). The initial stage was characterized by downregulation of meta-stable (MS) genes from the Pre-M state markers (enriched in the pathways of endodermal development) and upregulation of intermediate-hybrid (IH) genes (enriched in pathways of MAPK cascade and metabolic process) from the M state markers (Fig. 4i and Supplementary Table 4). This process by first losing En identity enables a conversion of Pre-M stable cells toward the transition cells. The second stage of the transition marked by the gradual down-regulation of TD genes mainly involves negative regulation of cardiac muscle cell differentiation and cardiac muscle tissue development (Fig. 4i and Supplementary Table 4). The final stage completes the transition process with the down-regulation of Pre-M state IH genes, along with up-regulation of MS genes (enriched in the cardiac muscle cell myoblast differentiation and outflow tract morphogenesis process) in the M state (Fig. 4i and Supplementary Table 4), making transition cells to finally convert into the mesodermal cells and establish the stable cell fate. The ordering of cells based on TCS has an overall increasing trend from Day 2 to Day 3 via the time point of Day 2.5 within the transition cells, corresponding to the noticed three-stage transition (Supplementary Fig. 8). Together, the transition cells locating near the saddle points connecting Pre-M (or Pre-En) and M (or En) reflect the temporal orderings of cell-fate conversion, which are well characterized by TD and IH genes in a system consisting of one pitchfork bifurcation.

**MuTrans reveals complex lineage dynamics in blood cell differentiation**. The hematopoiesis has been conceived as a hierarchy of discrete binary state-transitions, while increasing evidence alternatively supports a continuous and heterogeneous view of such process[38]. To investigate the complex dynamics in blood differentiation where transition cells likely play key roles, we applied MuTrans to different single-cell datasets with different sequencing depths and sample sizes.

We first analyzed the single-cell RNA data during myelopoiesis sequenced with Fluidigm C1 platform[39]. The number of attractors and cell label annotations are selected to recover the label resolutions in original publication. Notably MuTrans highlights the hub states—multi-lineage cells, which are capable of becoming three types of blood cells through a shallow basin resided in the highest terrain of the entire dynamical manifold (Fig. 5a and Supplementary Figs. 9–10). The low barriers between

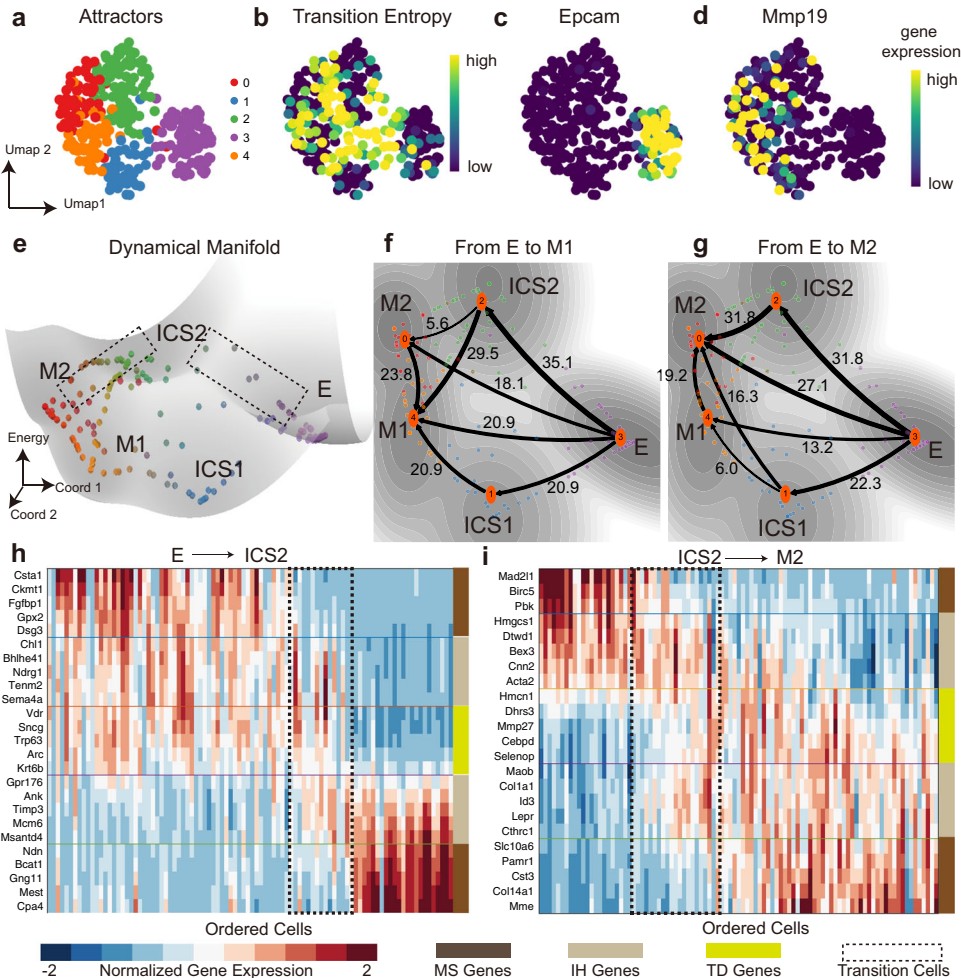

**Fig. 3 MuTrans reveals the cell-state transitions in EMT dataset of Squamous Cell Carcinoma. a** Attractor basins found by MuTrans, with cells plotted in the UMAP low-dimensional space and colored by MuTrans attractors. **b** Calculated transition entropy of each cell, with larger values corresponding to more intermediate cells with highly mixing identities. **c, d** Gene expression value of epithelial state (E) marker Epcam and mesenchymal state (M) marker Mmp19. **e** The dynamical manifold of EMT with cells colored by MuTrans attractors in (**a**). Two dashed squares indicate the transition processes presented in (**h**) and (**i**). Attractor 0 is annotated as the epithelial (E) state, attractor 0 and 4 as two mesenchymal (M) states, and attractor 1 and 2 as two intermediate cell states (ICS) of EMT. **f, g** The transition path analysis by setting E attractor as start state and two M attractors as target state, respectively, overlaid on the dynamical manifold. The numbers indicate the relative likelihood of each transition path. The cells are colored by MuTrans attractors. The grayness indicates energy values of dynamical manifold, with darker colors representing lower energy values. **h, i** Transition cells and genes analysis during the transitions from E to ICS2 (**h**) and from ICS2 to M2 (**i**). For each cell-state transition, the cells (columns) of heatmap are ordered by transition cell scores (TCS) of each process, and transition cells marked by the black dashed rectangles. The rows of heatmap are the top five most significant genes identified for the down-regulated MS (meta-stable) genes, down-regulated IH (intermediate-hybrid) genes, TD (transition-driver) genes, up-regulated IH genes and up-regulated MS genes, respectively (from top to bottom in heatmap).

the multi-lineage basin and the downstream basins (granulocytic or monocytic states) suggest probable transitions from the multi-lineage state, consistent with the observed transition cells across the saddle point. Interestingly, the transition cells during Multi-lin to Gran conversion were previously identified as the multi-lineage cells in ICGS clustering[39] (Supplementary Fig. 10). Similarly, during the megakaryocytic cell differentiation, while the transition cells consist of both HSPC1 and Meg types in our analysis, they were previously identified as the hematopoietic progenitor cells by the ICGS criterion (Supplementary Fig. 10). Such discrepancy could be explained by the gene expression dynamics in gradual transition of cell states. For example, during transition from multi-lineage cells to granulocytic cells (Fig. 5c), we observed the typical expression pattern of TD, MS and IH genes as conceptualized in Fig. 1e. Despite the similarity between the transition cells and their departing multi-lin state as manifested in the co-expression of down-regulated IH genes

(Fig. 5c, yellow lines), we also detected the up-regulated IH genes (Fig. 5c, yellow lines), suggesting the resemblance of transition cells with their targeting gran cell state (Supplementary Table 5). We observed a similar gene expression pattern in the transition from HSPC to Meg state (Supplementary Fig. 12 and Supplementary Table 6). For this dataset, MuTrans is able to capture the established attractor cell states, in addition to finding transition cells that were classified in some stable states by a previous study[39].

Focusing on the dataset of cell-fate bias toward lymphoid lineage, MuTrans resolves the complex lineage dynamics underlying single-cell RNA data of mouse hematopoietic progenitors differentiation sequenced from Cel-Seq2 platform[40]. Consistent with the major findings of FateID algorithm, the constructed dynamical manifold reveals that lymphoid progenitor (LP) cells (red balls) give rise to both B cells (pink balls) and plasmacytoid dendritic cells (pDCs) (Fig. 5b and Supplementary Fig. 13). The

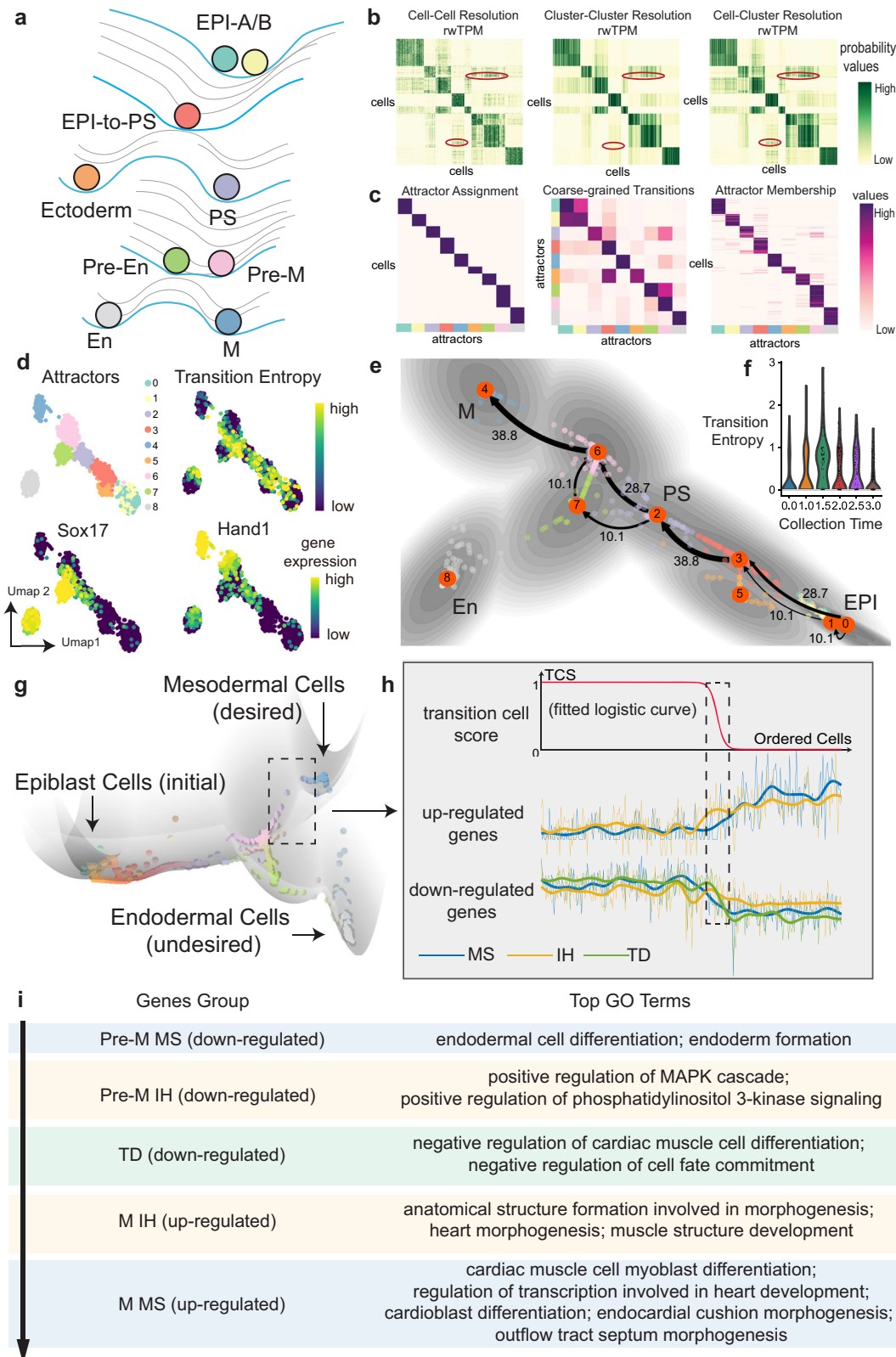

inferred dynamical manifold also suggests that certain transition cells in the attractors of pDCs originate directly from multi-potent progenitor (MPP) cells (yellow balls, Supplementary Fig. 13). MuTrans resolves the details in B cell differentiation, capturing the transition cells from Pro-B toward Pre-B basins (Supplementary Fig. 13 and Supplementary Table 7). Downstream analysis suggested the transition cell features by the co-expressed IH genes (yellow lines, Fig. 5d) and the dynamically expressed TD genes (green lines, Fig. 5d). Overall, MuTrans can provide a global cell-fate transition picture with marked transition cells in this dataset of highly complex lineages, in addition to the local transition routes inferred by FateID[40].

**Fig. 4 MuTrans scrutinizes the cellular bifurcation and gene expression dynamics during iPSC differentiation. a** The schematic development landscape during iPSCs differentiation, with cell states and lineage relationship inferred by MuTrans. EPI: epiblast cells. PS: primitive-streak cells. En: endodermal cells. M: mesenchymal cells. **b**, **c** The multi-scale quantities learned by MuTrans. **b** The learned cellular random walk transition probability matrix (rwTPM). Elements in red circle indicate that cell-cluster scale rwTPM recovers the finer resolution of cell-cell scale rwTPM than the cluster-cluster scale rwTPM. **c** The cell-cluster assignment, cluster-cluster transition probability and cell-cluster membership matrix learned by MuTrans. **d** MuTrans outputs shown in UMAP dimension reduction plot, including attractor basins, transition entropy of each cell and gene expression values of endodermal (En) marker Sox17 and mesodermal (M) marker Hand1. **e** The transition path analysis by setting epiblast (EPI) attractor as start state and the mesodermal (M) attractor as target state, overlaid on the two-dimensional dynamical manifold. The numbers indicate the relative likelihood of each transition path. The cells are colored by MuTrans attractors. The grayness indicates energy values of dynamical manifold, with darker colors representing lower energy values. **f** The violin plot of transition entropy distributions grouped and colored by different cell collection time points (days). **g** The constructed dynamical manifold. The color of each individual cell is computed based on the value of its soft clustering membership. **h** The Transcendental analysis of the transition from Pre-M state to M-state. The TCS (transition cell score) are shown with transition cells marked by dashed rectangles. Transition cells are marked by dashed squares. The average gene expression of top 5 MS (meta-stable genes, blue), IH (intermediate-hybrid genes, yellow) and TD (transition-driver genes, green) are displayed over the ordered cells in transitions. The full gene name list is shown in Supplementary Table 4. The thin lines represent the raw normalized expression value and thick lines denote the smoothed data. **i** GO (gene ontology) enrichment analysis of MS, IH and TD genes during Pre-M to M state transition indicates a gradual loss of endodermal property and gain of mesodermal property in the cell-fate switch.

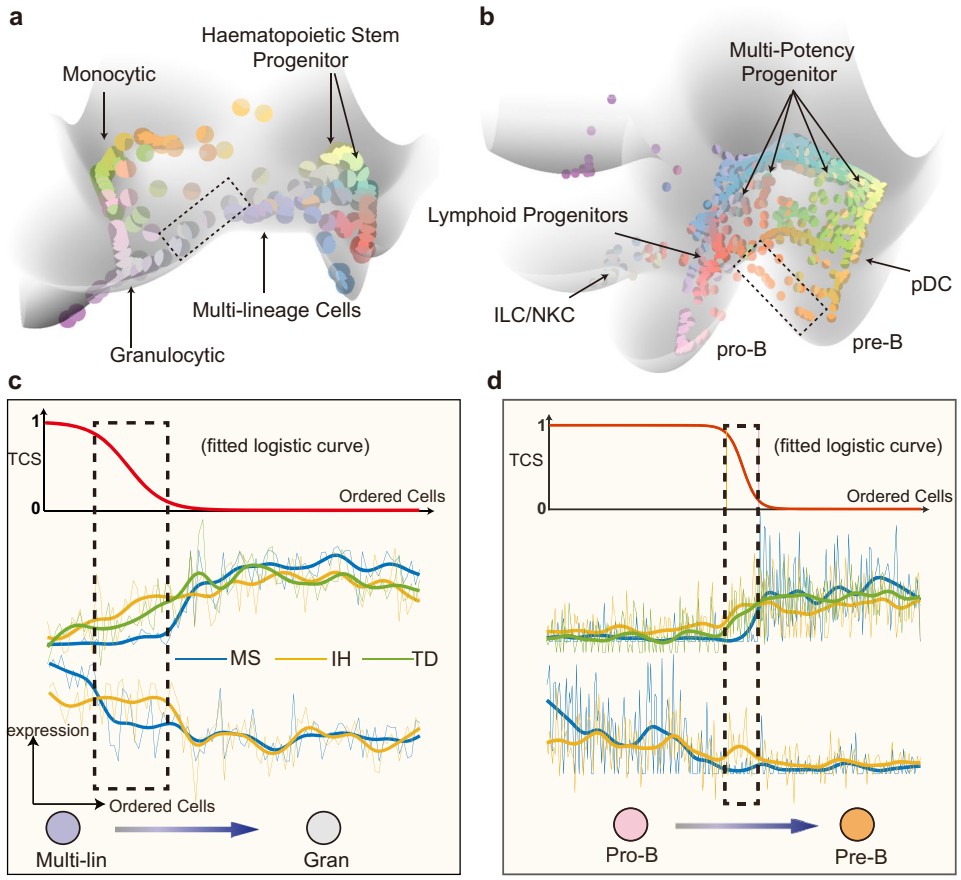

**Fig. 5 MuTrans can reveal the underlying complex dynamics in single-cell blood differentiation datasets. a**, **b** The constructed dynamical manifold by MuTrans is shown for the two datasets. The color of each individual cell in dynamical manifold is based on its soft-clustering membership value. In mouse blood cells dataset (**a**), MuTrans highlights the multi-lineage cells in a shallow pit on dynamical manifold. In the blood cells dataset toward lymphoid lineages (**b**), MuTrans discovers plenty of transition cells exist between stable Pro-B and Pre-B cell attractors (marked by dashed squares). pDC: plasmacytoid dendritic cells. ILC: innate lymphoid cells. NKC: natural-killer cells. The complete cell type annotations are shown in Supplementary Fig. 12. **c**, **d** The TCS (transition cell score) of transition and average gene expression of the top 5 TD (transition-driver, green), MS (meta-stable, blue) and IH (intermediate-hybrid, yellow) genes for the two interested transition paths marked with dash in (**a**, **b**). Transition cells are marked by dashed squares. The thin lines represent the raw normalized expression value and thick lines denote the smoothed data. The complete gene lists are shown in Supplementary Table 6–8. Multi-lin: multi-lineage cells. Gran: granulocytic cells.

**Application to large-scale datasets with complex trajectory**. To test the scalability of MuTrans, we studied on the single-cell hematopoietic differentiation data in human bone marrow generated by 10x Chromium platform[41] (Fig. 6a). To make the comparison, we applied MuTrans to both the complete (original)

data, and the one after using the pre-processing module DECLARE. We found DECLARE could reduce the calculation time by one magnitude for this dataset.

For both cases MuTrans identified the expected bifurcations from hematopoietic stem cells (HSC) into the monocytic

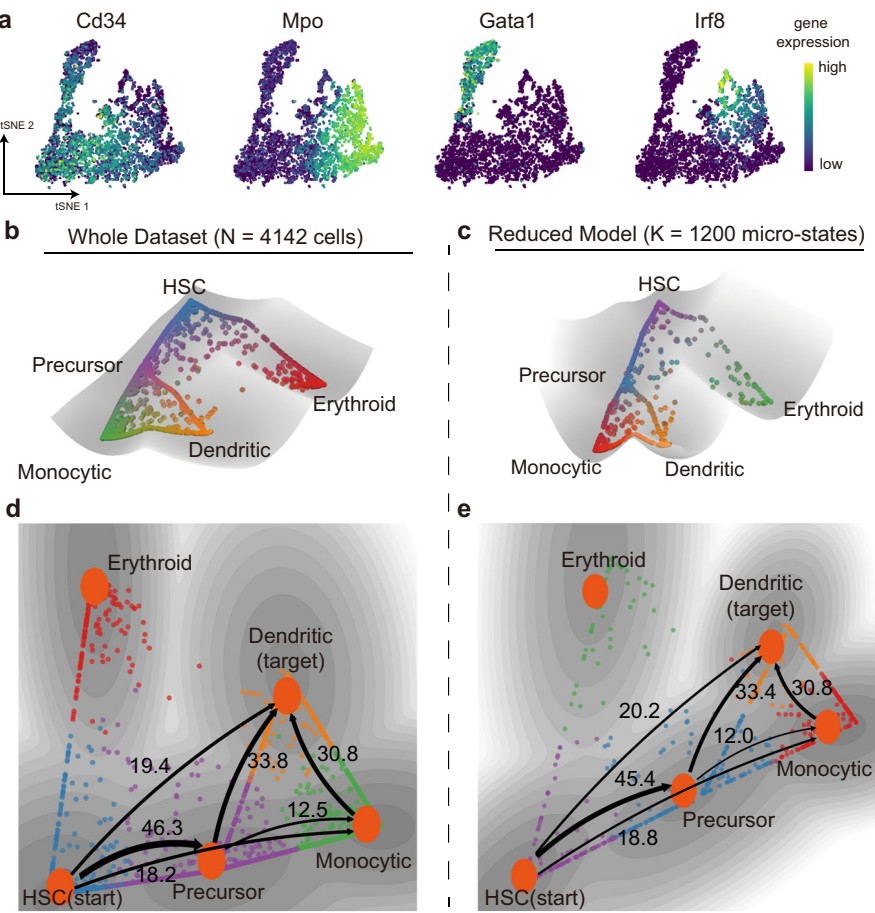

**Fig. 6 Application to a large dataset using multiscale reduction approach. a** The tSNE plot and marker gene expression of datasets from early human HSC differentiation in bone marrow. **b, c** The dynamical manifold constructed from complete dataset (**b**, $N = 4{,}142$ cells) and with DECLARE pre-processing (**c**, $K = 1{,}200$ micro-states) with cells colored by soft clustering membership in MuTrans attractors. In (**b**) each ball represents one cell and in (**c**) each ball represents one micro-state. The reduced model preserves the overall structure of dynamical manifold. HSC: hematopoietic stem cells. **d, e** The transition paths analysis conducted on complete data (**d**) and with DECLARE pre-processing (**e**), where HSC are picked as the start and dendritic cells as the target. The numbers indicate the relative likelihood of each transition path, suggesting the quantitative consistency of reduced model with the analysis on whole dataset. Cells are colored by MuTrans attractors. The grayness indicates energy values of dynamical manifold, with darker color representing lower energy values.

precursors and erythroid cells, as well as the differentiation from precursor cells into monocytic and dendritic cells. The constructed dynamical manifold (Fig. 6b, c and Supplementary Fig. 14) shows a continuous stream of transition cells among different basins (such as those moving between dendritic and monocytic potential wells) suggesting the hematopoietic differentiation may be a continuous process. The transition trajectories obtained with the large-scale pre-processing step are consistent with the complete dataset analysis (Fig. 6d, e). This indicates the major transition trajectories toward dendritic cell fate not only consist of the path mediated by monocytic precursor states but also include a considerable flux of transition cells from differentiated monocytic cells. Interestingly, the existence of both stable states and transition cells reconciles a previously noted discrepancy[41] caused by treating the underlying cellular transition dynamics as either a purely continuous processing (e.g., using Palantir) or a discrete process (using other clustering-based lineage inference methods such as Slingshot[14] and PAGA[42]).

Next, we analyzed another dataset containing over 15,000 cells collected during blood emergence in mouse gastrulation[43] (Fig. 7a). Consistent with the PAGA[42] low-dimensional embedding of the data (Fig. 7b), the constructed dynamical manifold (Fig. 7c) and

derived Maximum Probability Flow Tree (MPFT) suggest three major transition branches from haemato-endothelial (Haem) cells into endothelial cells (EC), mesoderm cells (Mes) or erythroid cells (Ery). Specifically, the transition path analysis indicates that the endothelial cells and erythroid cells are originated through discrete trajectories from haemogenic endothelium (Fig. 7e), and such trajectories are mediated by the intermediate state of blood progenitor (BP) cells (Fig. 7f). These results are consistent with the experimental findings on endothelial and erythroid cells[43].

**Comparison and consistency with other methods**. MuTrans is designed specifically to identify transition cells, with its theory rooted in multi-scale dynamical systems and allowing natural visualization and quantification of cell-state transitions. To compare with other methods which may provide information on transitions, we performed further analysis with pseudo-time ordering and cell-fate bias probability methods on their capability of detecting transition cells, using existing methods, such as PAGA, FateID and VarID (Supplementary Note 4).

In iPSC data, we found that MuTrans, PAGA and VarID are consistent in recovering the bifurcation dynamics toward En and M states (Supplementary Fig. 15). While the projected lineage tree of StemID2 shows transition cells between precursor and mature

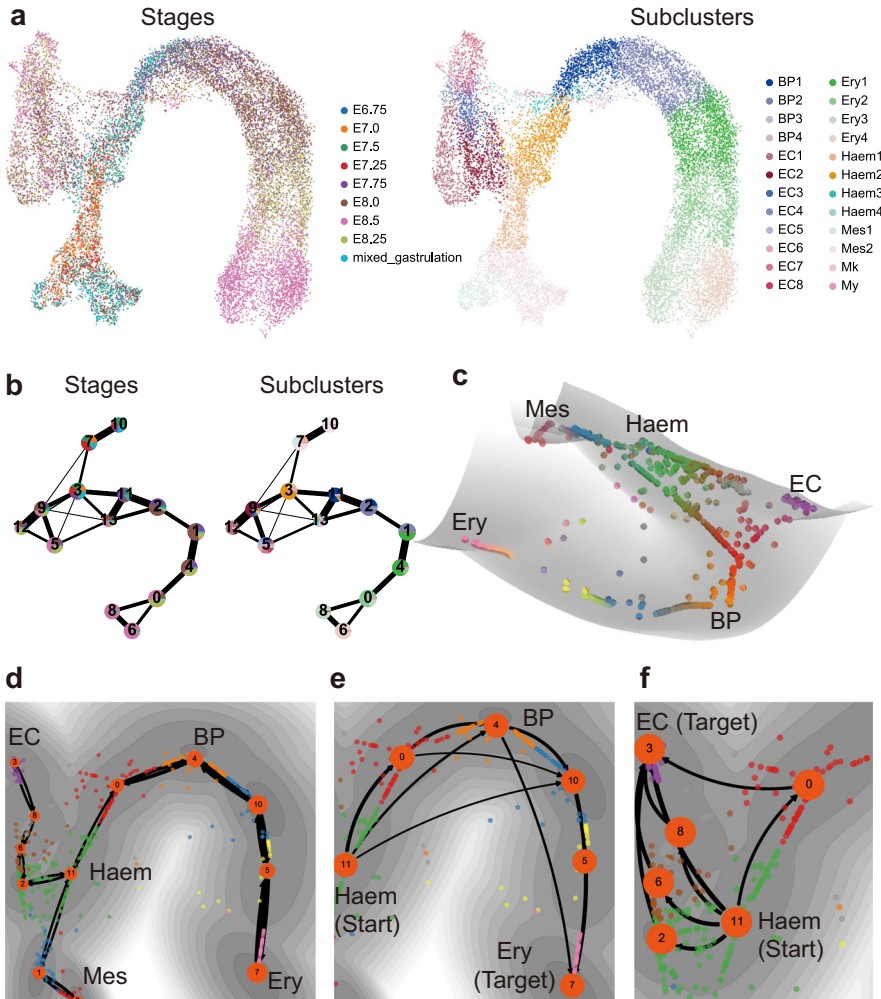

**Fig. 7 Application to a dataset on blood cell differentiation in mouse gastrulation ($N = 15,875$ cells). a** The UMAP plot with cells colored by experimental collection time and the cell annotations in original publication. BP: blood progenitor cells. EC: endothelial cells. Ery: erythrocytic cells. Haem: haemato-endothelial cells. Mes: mesoderm cells. Mk: megakaryocytic cells. My: myeloid cells. **b** The cell lineage inferred by PAGA with the coarse-grained states colored by experimental collection time and the cell annotations in the original study. **c** The dynamical manifold constructed by MuTrans with DECLARE pre-processing ($K = 1,500$ micro-states), with cells colored by soft clustering membership in MuTrans attractors. **d** The global cell lineage inferred by MuTrans MPFT (maximum probability flow tree) algorithm, overlaid on the dynamical manifold. Cells are colored by MuTrans attractors. The grayness indicates energy values of dynamical manifold, with darker color representing lower energy values. **e** Zoom-in of the dominant transition paths from Haem cells to endothelial cells. **f** Zoom-in of the dominant transition paths from Haem cells to erythrocytic cells.

En/M states (Supplementary Fig. 15), the reconstructed spanning tree does not reveal the overall bifurcation structure.

For the myelopoiesis dataset, we found that both MuTrans and VarID recover the bifurcations toward granulocytic and monocytic states (Supplementary Fig. 16). Consistent with MuTrans, FateID also captures the differentiation paths toward monocytic states (Supplementary Fig. 16).

Close inspection into the transition from precursors to mature En/M states in iPSC dataset suggests that based on existing approaches (such as tracking the changes along pseudotime or fate bias probability) could not distinguish the transition cells from stable cells as accurately and reliably as MuTrans. Both Monocle3 and DPT have a sharp increase in the pseudotime during the transitions (Supplementary Fig. 17), therefore lacking resolution in probing the transition cells linking multiple attractors. Fate ID suggests a gradual change of En/M fate probability in precursor cells (Supplementary Fig. 17), not discriminating the transition cells within Pre-En and Pre-M states. Such problem was also observed when using Palantir,

which depicts the entire cell-state transition as a highly continuous and gradual process (Supplementary Fig. 17).

## Discussion

Overall, MuTrans provides a unified approach to inspect cellular dynamics and to identify transition cells directly from single-cell transcriptome data across multiple scales. Central to the method is an underlying stochastic dynamical system that naturally connects (1) attractor basins with stable cell states, (2) saddle points with transient states, and (3) most probable paths with cell lineages. Instead of the widely used low-dimensional geometrical manifold approximation for the high-dimensional single-cell data, our method constructs a cell-fate dynamical manifold to visualize dynamics of cells development, allowing direct characterization of transition cells that move across barriers amid different attractor basins. Adopting the transition path theory to the multiscale dynamical system, we quantify the relative likelihoods of various transition trajectories that connect a chosen

root state and the target states. In addition, we provide a quantitative methodology to detect critical genes that drive transitions or mark stable cells.

In this study a key theoretical assumption for modeling cell-state transition is a barrier-crossing picture in multi-stable dynamical systems, a concept which has been adopted for describing cell developments through dynamical system language[3,44,45]. Indeed, the notions of barriers, saddles and potential landscape underlying the actual biological process are the emergent properties of the complex interactions, such as gene expression regulation and signal transduction during a developmental process[28]. The driving force that overcomes the barrier and induces the transition may arise from both the extrinsic environment and the fluctuations within the cells[46]. Multiscale reductions used by MuTrans naturally capture the transition cells, allowing inference of the corresponding transition processes.

Pseudo-time ordering and low-dimensional trajectory embedding may serve as intuitive tools to trace the progression of cell fates by comparing similarity of gene expression among cells. Such approaches often adopt the deterministic point of view and rely on the low-dimensional projection of datasets, lacking theoretical insights to the underlying dynamical processes of cell-state transitions. In contrast, MuTrans is based on multi-stable dynamical system approach in characterizing cell-state transitions. While cells reside and fluctuate within attractor basins for majority of time, it is the temporal ordering of transition cells, rather than stable cells, reflect the actual process of cell transitions (Fig. 1c and Supplementary Fig. 17).

Methods such as Palantir[41], Population Balance Analysis (PBA)[29] and Topographer[47] also treat cell-fate transition as Markov random walk process. These methods depict the dynamics at the individual cell level, then compute pseudo-time ordering based on the first passage time or absorbing probabilities of the Markov Chain. In comparison, MuTrans can dissect the intrinsic multiscale features of the system and derive the coarse-grained dynamics, distinguish between stable and transition cells quantitatively, and characterize multiple and complex routes of transition paths.

Several other methods[2,48] define the transition probabilities between clusters based on entropy difference or summing up the cell-cell transition probabilities. Here the coarse-grained transition probability in MuTrans is an emergent quantity derived from multiscale reduction. The transition probability is shown to be consistent with Kramers' reaction rate theory for over-damped Langevin dynamics if steady-state assumption and detailed-balanced condition are satisfied (Methods and Supplementary Note 4).

To describe the smooth state transitions, some methods[49,50] adopt the soft-clustering strategy based on the soft K-means or factor decomposition for gene expression matrix. In comparison, the soft cell assignment of MuTrans is obtained from multiscale learning of cell-cluster rwTPM, which can be more robust against technical noise than using gene expression matrix directly for clustering[7]. Such robustness is critical to detecting transition cells in datasets with lower sequencing depth, such as 10X data. Beyond interpreting the soft membership function as the indicator of cell locations in attractor basins, it remains an interesting problem to derive its continuum limit in the embedded over-damped Langevin dynamical systems.

To deal with the emerging large-scale scRNA-seq datasets, MuTrans introduces a pre-processing method (DECLARE) to aggregate the cells and speed up computation. The aggregation method uses the coarse-grain approach consistent with MuTrans, and it is different from other methods often used for large scRNA-seq datasets, such as down-sampling convolution[51] or kNN partition[52] that is based on the averaging or summation of cells with similar gene expression profiles. As a result, DECLARE can be naturally integrated with dynamical manifold construction and transition trajectory inference.

The stochastic transitions among attractors considered by MuTrans can be further incorporated with deterministic processes to better understand the cell-fate decision[53]. Despite that the stochastic switching among cell states might be rare in some cases, the local fluctuation of microscopic cell states in gene expression can be prevalent in the microscopic dynamics, therefore the cell-cell scale random walk assumption in MuTrans still holds as a natural assumption. In theory, the stochastic transition model is consistent with the uni-directional transition process if the transition probabilities in one direction are dominant, or when the noise amplitude of system is relatively small.

The theoretical assumptions on equilibrium and steady-state systems made in MuTrans can be potentially mediated by our multiscale approach. For example, although the detailed balance may be violated at the microscopic scale described in Eq. (1) the estimated coarse-grained (mesoscopic) dynamics in MuTrans can be sufficient to recover the transitions at larger scale. However, non-stationary effects due to cell cycle or cell proliferation dynamics[29] were not considered in current method. In addition, the number of cells in the datasets, in principle, needs to be sufficiently large in order to obtain high-resolution identification of transition cells. When the number of cells is relatively small, such as in the myelopoiesis dataset studied here, special care is needed to further confirm the analysis of transition cells. Besides, more effective ways in root cell states detection (e.g., through entropy methods[54] or RNA velocity[55–57]) can further enhance the robustness of our approach.

In addition to infer complex cellular dynamics induced by transition cells from single-cell transcriptome data, MuTrans along with its computational or theoretical components can be used for development of other approaches for dissecting cell-fate transitions from both data-driven and model-based perspectives.

## Methods

MuTrans performs three major tasks in order to reveal the dynamics underneath single-cell transcriptome data (Fig. 1): 1) assigning each cell in the attractor basins of an underlining dynamical system, 2) quantifying the barrier heights across the attractor basins, and 3) identifying relative positions of the cells within each attractor. The first two tasks are executed simultaneously through the coarse-graining of multi-scale cellular random walks, an alternative approach to the traditional clustering of cells and inference of cell lineage. The third task is achieved by refining the coarse-grained dynamics via soft clustering, and serves as a critical procedure to identifying the transition cells during cell-fate conversion.

### Multi-scale analysis of the random-walk transition probability matrix (rwTPM). We assume the underlying stochastic dynamics during cell-fate conversion is modeled by random walks among individual cells through the random-walk transition probability matrix (rwTPM). Dependent on the choices of either cell-level or cluster-level, the rwTPM can be constructed in different resolutions, exhibiting multi-scale property and leading the identification of transition cells from the stable cells.

In describing the method, we use the indices $x, y, z$ to denote individual cells and $i, j, k$ to represents the clusters (or cell states) for the simplicity of notations.

1. The rwTPM in the cell-cell resolution
   The rwTPM $\mathbf{p}$ of cellular stochastic transition can be directly constructed from the gene expression matrix in cell-cell resolution, with the form

$$p(x, y) = \frac{w(x, y)}{d(x)}, d(x) = \sum_z w(x, z) \tag{2}$$

where the weight $w(x, y)$ denotes the affinity of gene expression profile in cell $x$ and $y$ (Supplementary Note 2). Such microscopic random walk yields an equilibrium probability distribution $\mu(x) = \frac{d(x)}{\sum_z d(z)}$, satisfying the detailed-balance condition $\mu(x)p(x, y) = \mu(y)p(y, x)$. The rwTPM captures the cellular transition in the cell-cell resolution (Fig. 1d).

2. The rwTPM in the cluster-cluster resolution
   The cellular transition rwTPM can be lifted in the cluster-cluster resolution by adopting a macroscopic perspective. For example, the cell-to-cell rwTPM can be generated from certain coarse-grained dynamics, by assigning each cell in different attractors $S = \bigcup_{k=1}^{K} S_k$, and model the transitions as the

Markov Chain among attractors with the transition probability matrix $\hat{\mathbf{P}} = (\hat{P}_{ij})_{K \times K}$. Here $\hat{P}_{ij}$ denote the probability that the cells reside in the attractor $S_i$ switch to the attractor $S_j$. The number of attractors $K$ is a hyperparameter of algorithm selected by the user. We use the Eigen-Peak Index (EPI) to visualize the multiple eigen-gaps of cell-cell scale rwTPM (Supplementary Note 2). Different peaks in EPI correspond to the number of attractors in different resolutions. In practice, the choice of $K$ can also be determined based on prior biological knowledge such as marker genes expression or known cell-type annotations.

Denote $1_{S_k}(z)$ as the indicator function of cluster $S_k$ such that $1_{S_k}(z) = 1$ for cell $z \in S_k$ and $1_{S_k}(z) = 0$ otherwise. The cluster-cluster transition based on probability matrix $\hat{\mathbf{P}}$ can naturally induce another rwTPM $\hat{\mathbf{p}}$ with the form

$$\hat{p}(x, y) = \sum_{i,j} 1_{S_i}(x) \hat{P}_{ij} 1_{S_j}(y) \frac{\mu(y)}{\hat{\mu}_j}, \tag{3}$$

where $\hat{\mu}_j = \sum_y 1_{S_j}(y) \mu(y)$ is the stationary probability distribution of cluster $S_j$. Intuitively, the stochastic transition from cell $x \in S_i$ to $y \in S_j$ can be decomposed into a two-stage process: a cell switches cellular state from cluster $S_i$ to $S_j$ with probability $\hat{P}_{ij}$, and then becomes the cell $y$ in cluster $S_j$ according to its relative portion at equilibrium $\frac{\mu(y)}{\hat{\mu}_j}$. The rwTPM captures the cellular transition in the cluster-cluster resolution (Fig. 1d).

3. The rwTPM in the cell-cluster resolution

Because some cells, for example the transition cells, may not be characterized by their locations in one basin, we introduce a membership function $\rho(x) = (\rho_1(x), \rho_2(x), \dots, \rho_K(x))^T$ for each cell $x$ to quantify its uncertainty in clustering. The element $\rho_k(x)$ represents the probability that the cell $x$ belongs to cluster $S_k^*$ with $\sum_k \rho_k(x) = 1$. For the cell possessing mixed cluster identities, its membership function $\rho(x)$ might have several significant positive components, suggesting its potential origin and destination during the transition process. In terms of dynamical system interpretation, the membership function captures the finite-noise effect in over-damped Langevin equation, which introduces the uncertainty of transition paths across saddle points[58], revealing that cells near saddle points and stable points may exhibit different behaviors in the state-transition dynamics.

From the coarse-grained dynamics $\left(\{S_k\}_{k=1}^K, \{\hat{P}_{ij}\}_{i,j=1}^K\right)$ and the measurement of cell identity uncertainty $\rho_k(x)$ in the clusters, one can reinterpret the induced microscopic random walk $\tilde{\mathbf{p}}$ in a cell-cluster resolution as

$$\tilde{p}(x, y) = \sum_{i,j} \rho_i(x) \hat{P}_{ij} \rho_j(y) \frac{\mu(y)}{\tilde{\mu}_j}, \tilde{\mu}_j = \sum_x \rho_j(x) \mu(x), \tag{4}$$

in parallel to Eq. (3) Now the transition from cell $x$ to $y$ is realized in all the possible channels from attractor basin $S_i$ to $S_j$ with the probability $\rho_i(x) \rho_j(y)$. The underlying rationale is that the transition can be decomposed in a three-stage process: First we pick up cell starting in attractor basin with membership probability, then conduct the transition with coarse-grained probability between attractor basins, and finalize the process by picking the target cell with membership probability in the target attractor basin. Now the rwTPM captures cellular transition in the cell-cluster resolution (Fig. 1d).

4. Integrating the rwTPM at three levels

To integrate the rwTPM from different resolutions, we next optimize the rwTPM on cluster-cluster and cell-cluster level through approximating the original rwTPM in the cell-cell resolution. First, we seek an optimal coarse-grained reduction that minimizes the distance between $\hat{\mathbf{p}}[S_k, \hat{P}_{ij}]$ and $\mathbf{p}$ by solving an optimization problem:

$$\min_{S_k, \hat{P}_{ij}} \mathcal{J}[S_k, \hat{P}_{ij}] = \|\hat{\mathbf{p}}[S_k, \hat{P}_{ij}] - \mathbf{p}\|_\mu^2, \tag{5}$$

where $\mu$ is the stationary distribution of original cell-cell random walk $\mathbf{p}$, and $\|\cdot\|_\mu$ is the Hilbert-Schmidt norm[59] for given transition probability matrix $\mathbf{A}$, defined as $\|\mathbf{A}\|_\mu^2 = \sum_{x,y} \frac{\mu(x)}{\mu(y)} A(x, y)^2$. The optimization problem is solved via an iteration scheme for $S_k$ and $\hat{P}_{ij}$ respectively (Supplementary Note 2). The optimal coarse-grained approximation $\left(S_k^*, \hat{P}_{ij}^*\right)$ indicates the distinct clusters of cells and their mutual conversion probability. Provided with the starting state, we can infer the cell lineage from the Most Probable Path Tree (MPPT) approach or Maximum Probability Flow Tree (MPFT) approach (Supplementary Note 2).

Next, we optimize the membership $\rho_k(x)$ such that the distance between the cell-cluster rwTPM $\tilde{\mathbf{p}}$ and the original $\mathbf{p}$ is minimized, i.e.,

$$\min_{\rho_k} \mathcal{E}[\rho_k] = \|\tilde{\mathbf{p}}[\rho_k] - \mathbf{p}\|_\mu^2 \tag{6}$$

$$\text{s.t.} \sum_k \rho_k(x) = 1, \rho_k(x) \geq 0 \text{ for } k = 1, .., K \text{ and } x \in S$$

with the initial condition $\rho_i^0(x) = 1_{S_i^*}(x)$, and $\tilde{\mathbf{p}}[\rho_k]$ is defined from Eq. (4) by plugging in the obtained $\hat{P}_{ij}^*$. The optimization problem is solved by the quasi-Newton method (Supplementary Note 2). The obtained membership function

$\rho^*(x)$ specifies the relative position of the cells within each attractor basin and is optimal in the sense that it guarantees the closest approximation of cell-cluster level rwTPM toward the cell-cell level transition dynamics.

**Transition entropy**. To quantify and compare the transition cells around different attractors in a global view, we define a transition entropy $H(x)$ for each cell $x$ based on the obtained membership function $\rho^*(x)$,

$$H(x) = -\sum_{k=1}^K \rho_k^*(x) \log \rho_k^*(x). \tag{7}$$

According to the definition, a stable cell tends to have a relatively small entropy value close to zero, while a transition cell, which possesses multiple and more evenly distributed components in its membership function, tends to have a larger transition entropy. As a result, a large entropy value indicates a cell with highly mixing identity, a case for transition cells in bifurcating attractors. The increase of transition entropy value can be utilized as a way to mark cell-state bifurcations.

**Transition paths quantification and comparison**. To quantify the cell development routes, we use the transition path theory based on coarse-grained dynamics $\left(\{S_k\}_{k=1}^K, \{\hat{P}_{ij}\}_{i,j=1}^K\right)$ to compare the likelihood of all possible transition trajectories. Given the set of starting states $A$ and the targeting state $B$, we calculate the effective current $f_{ij}^+$ of transition paths passing through state $S_i$ to $S_j$ based on the inferred attractor basins and conversion probabilities (Supplementary Note 2), and specify the capacity of given development route $w_{dr} = (S_{i_0}, S_{i_1}, .., S_{i_n})$ connecting sets $A$ and $B$ as $c(w_{dr}) = \min_{0 \leq k \leq n-1} f_{i_k i_{k+1}}^+$. The likelihood of transition trajectory $w_{dr}$ is defined as the proportion of its capacity to the sum of all possible trajectory capacities. In the python package of MuTrans, we use the functions in PyEMMA[60] for the computations.

**Pre-processing by DECLARE and scalability to large datasets**. To reduce the computational cost for large datasets (for instance, greater than 10 K cells), we introduce a pre-processing module DECLARE (dynamics-preserving cell aggregation). The module first detects the hundreds/thousands of microscopic attractor states by clustering (e.g., using K-means or kNN partition) and then derive the coarse-grained transition probabilities among these microscopic attractor states. Based on such transition probabilities, we then follow the standard multiscale reduction procedure of MuTrans to find macroscopic attractor states, construct dynamical manifold, quantify the transition trajectories and highlight the transition states (Supplementary Note 2).

**Transition cells and genes analysis through transcendental**. Based on the soft clustering results, MuTrans performs the Transcendental (transition cells and relevant analysis) procedure on each transition process to identify the transition cells from the stable cells and reveal the relevant marker genes.

For the given transition process from attractors $S_i^*$ to $S_j^*$ along the transition path, we first selected the cells relevant to the transition, based on the membership function $\rho^*(x)$ (Supplementary Note 2). Then for each relevant cell $x$, we define the transition cell score (TCS)

$$\tau_{ij}(x) = \frac{\rho_i^*(x)}{\rho_i^*(x) + \rho_j^*(x)}, \tag{8}$$

to measure the relative position of cell $x$ in different clusters. Here the TCS $\tau_{ij}$ takes the values near zero or one when a cell resides around the attractor in $S_i^*$ or $S_j^*$ (i.e., the cells are stable), whereas yields the intermediate value between zero and one for the cell that possesses a hybrid or transient identity of two or more clusters. Next we arrange all the relevant cells in state $S_i^*$ and $S_j^*$ according to $\tau_{ij}$ in descending order, and the reordered $\tau_{ij}$ indicates a sharp transition (Fig. 1a) or a smooth transition (Fig. 1a) from the value one to zero. For the smooth transition, there is a group of cells whose value of $\tau_{ij}$ decreases gradually from one to zero (Fig. 1e). This group of cells in the transition layer are called the transition cells from state $S_i^*$ to state $S_j^*$, and their order reflects the details of the state-transition process. To quantify the transition steepness, we use logistic functions to model the transition and estimate the relative abundance of transition cells (Supplementary Note 2).

Differentially expressed genes analysis is usually applicable when the clusters are distinct and the state-transition is sharp (Fig. 1a). However, to characterize the dynamical and hybrid gene expression profiles in transition cells, merely comparing the average gene expression in different clusters is insufficient. Here we define three kinds of genes relevant to the state transition of cells: a) the transition-driver (TD) genes that vary accordingly with the transition dynamics, b) the intermediate-hybrid (IH) genes marking the hybrid features from multiple cell states that are expressed in the intermediate transition cells, and c) the meta-stable (MS) genes that represent cells in the stable states.

The expression of TD genes varies accordingly to the transition, revealing the driving mechanism of the cell-state conversion. To probe TD genes, we calculate the correlation between the gene expression values and $\tau_{ij}$ in the ordered transition

cells. The genes with larger correlation values (larger than a given threshold value) are identified as TD genes. The IH genes express eminently both in the transition cells and in the stable cells from one specific cluster, reflecting the hybrid state of the transition cells, while the MS genes express exclusively in the stable cells from certain cluster. To distinguish IH and MS genes from all the differentially expressed genes, we compare the gene expression values between the stable cells and the transition cells, respectively, within each cluster. The significantly up-regulated genes in the stable cells are defined as the MS genes, and the rest differentially expressed genes are identified as the IH genes that express simultaneously both in stable and transition cells (Supplementary Note 2). Here the selected genes only reflect the relative gene expression trends amid one specific cell-state transition process, without considering global comparisons between multiple cell states or transitions. Therefore, the MS genes, which distinguish the attractors and transition cells locally in the dynamical manifold, can be different from the conventional marker genes that are uniquely and strongly expressed in one cell state. Together with IH and TD genes, they provide useful information to identify genes that are driving the local transition.

**Constructing the cell-fate dynamical manifold.** To better visualize the transition process and their connections with cell states, MuTrans introduces the dynamical manifold concept. The construction of the dynamical manifold consists of two steps: (1) locating the center positions of cell clusters (corresponding to the attractors) in low dimensional space, (2) assigning the position of each individual cells according to soft-clustering membership function.

The initial center-determination step starts with an appropriate two-dimensional representation, denoted as $\mathbf{x}^{2D}$ for each cell $x$ (Supplementary Note 2). Instead of directly utilizing $\mathbf{x}^{2D}$ as the cell coordinate, we calculate the center $\mathbf{Y}_k$ of each cluster $\{S_k^b\}_{k=1}^K$ by taking the average of $\mathbf{x}^{2D}$ over cells within certain range of cluster membership function $\rho_k^*(x)$. Having determined the position of attractors, we define a two-dimensional embedding $\boldsymbol{\xi}(x)$ for each cell according to the membership function $\rho^*(x)$, such that $\boldsymbol{\xi}(x) = \sum_k \rho_k^*(x)\mathbf{Y}_k \in \mathbb{R}^2$. For the cell possessing mixed identities of state $S_i^*$ and $S_j^*$, its transition coordinate then lies in a value between $\mathbf{Y}_i$ and $\mathbf{Y}_j$.

For Fokker-Planck equation of the over-damped Langevin equation, the expansion of steady-state solution near stable points (attractors) indeed yields a Gaussian-mixture distribution[61]. Motivated by this, to obtain the global dynamical manifold we fit a Gaussian mixture model with a mixture weight $\hat{\boldsymbol{\mu}}^*$ to obtain the stationary distribution of coarse-grained dynamics. The probability distribution function of the mixture model becomes

$$\wp(\mathbf{z}) = \sum_k \hat{\mu}_k^* \mathcal{N}(z; \mathbf{Y}_k, \boldsymbol{\Lambda}_k), \tag{9}$$

where $\mathcal{N}(z; \mathbf{Y}_k, \boldsymbol{\Lambda}_k)$ is a two-dimension Gaussian probability distribution density function with mean $\mathbf{Y}_k$ and covariance $\boldsymbol{\Lambda}_k$. The landscape function of dynamical manifold is then naturally takes the form in two dimensions $\varphi(z) = -\ln\wp(z)$. Specifically, the energy of individual cell $x$ is calculated as $\varphi(\boldsymbol{\xi}(x))$. The constructed landscape function captures the multi-scale stochastic dynamics of cell-fate transition, by allowing typical cells that are distinctive to certain cell states positioned in the basin around corresponding attractors, while the transition cells laid along the connecting path between attractors across the saddle point. Moreover, the relative depth of the attractor basin reflects the stationary distribution of coarse-grained dynamics, depicting the relative stability of the cell states. The flatness of the attractor basin also reveals the abundance and distribution of transition cells, indicating the sharpness of cell fate switch. Theoretically, the constructed dynamical manifold approximates the energy landscape or quasi-potential[30,44,45] of underlying stochastic dynamical system.

**Mathematical analysis of MuTrans.** With the assumption that the single-cell data is collected from the probability distribution $v(x)$ with density of Boltzmann-Gibbs form, i.e., $\nu(x) \propto e^{-\frac{U(x)}{\varepsilon}}$, we can prove (Supplementary Note 1) that the microscopic random walk constructed by MuTrans can approximate the dynamics of over-damped Langevin Equation (OLE)

$$d\mathbf{X}_t = -\nabla U(\mathbf{X}_t)dt + \sqrt{2\varepsilon}d\mathbf{W}_t \tag{10}$$

in the limiting scheme, and the coarse-graining of MuTrans $(S_k, \hat{P}_{ij})$ is equivalent to the model reduction of OLE by Kramers' rate formula in the small noise regime, i.e., $k_{ij} \propto e^{-\frac{\Delta U}{\varepsilon}}$ as $\varepsilon \to 0$, where $k_{ij}$ is the switch rate from attractor $S_i$ to $S_j$, and $\Delta U$ denotes the corresponding barrier height of transition - the energy difference between saddle point and the departing attractor.

Therefore, if the cell transition dynamics can be well-modelled by the OLE dynamics of Eq. (10) MuTrans is indeed the multi-scale model reduction via the data-driven approach. In addition, the dynamical manifold constructed by MuTrans can be viewed as the data realization of potential landscape[44] for diffusion process in biochemical modelling, which incorporates the dynamical clues about the underlying stochastic system regarding the stationary distribution and transition barrier heights.

**Data simulation and analysis.** The simulation data was generated by the Euler-Maruyama method to solve the overdamped Langevin equations, with the detailed models and parameters specified in Supplementary Note 3.

The single-cell datasets analyzed were from different systems and platforms, namely mouse cancer EMT data (Smart-Seq2), mouse myelopoiesis data (Fluidigm C1), mouse hematopoietic progenitors data (Cel-Seq2), human hematopoietic progenitors data (10X Chromium), blood differentiation data (10X Chromium) in mouse gastrulation and iPSC induction data (single-cell RT-qPCR), downloaded from sources provided in Data availability section below. The detailed analysis for each dataset was provided in Supplementary Note 3. The full scripts for reproducing data analysis in main text and Supplementary Information for all the datasets are uploaded at https://github.com/cliffzhou92/MuTrans-release/tree/main/Example, with the processed gene expression matrices that could be loaded directly in MuTrans analysis stored at https://github.com/cliffzhou92/MuTrans-release/tree/main/Data.

We compared MuTrans with existing lineage inference methods Monocle 3[62], Diffusion Pseudotime[7], PAGA[42], FateID[40], RaceID 3 and StemID 2[40], VarID[48], Palantir[41] and PBA[29], with detailed settings for each method provided in Supplementary Note 4.

**Reporting summary.** Further information on research design is available in the Nature Research Reporting Summary linked to this article.

## Data availability

All the datasets used in this paper are publicly available. The mouse cancer EMT data (Smart-Seq2) used in this study was downloaded from the Gene Expression Omnibus (GEO) with accession number GSE110357. The mouse myelopoiesis data (Fluidigm C1) used in this study was downloaded from the Gene Expression Omnibus (GEO) with accession number GSE70245. The mouse hematopoietic progenitors data (Cel-Seq2) used in this study was downloaded from the Gene Expression Omnibus (GEO) with accession number GSE100037. The processed human hematopoietic progenitors data (10X Chromium) used in this study was downloaded from https://github.com/dpeerlab/Palantir/blob/master/data/marrow_sample_scseq_counts.csv.gz and processed blood differentiation data (10X Chromium) in mouse gastrulation used in this study was downloaded from https://github.com/MarioniLab/EmbryoTimecourse2018. The iPSC differentiation data (single-cell RT-qPCR) used in this study was downloaded from https://www.pnas.org/highwire/filestream/29285/field_highwire_adjunct_files/1/pnas.1621412114.sd02.xlsx. The codes and trajectories for simulation data, the processed single-cell data expression matrix, the MuTrans package and scripts to reproduce the figures and results in main text and repeat the detailed analysis in SI are also available at Github (https://github.com/cliffzhou92/MuTrans-release).

## Code availability

The Matlab implementation of MuTrans and affiliated Transcendental packages are available from GitHub (https://github.com/cliffzhou92/MuTrans-release). The Python package for MuTrans (pyMuTrans) compatible with Scanpy package[63] is also available in the repository.

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

## Acknowledgements
This project was supported by grants from the National Natural Science Foundation of China (11825102 and 11421101 to T.L.), National Institutes of Health grant U01AR073159 (Q.N.), National Science Foundation grants DMS1763272 (Q.N.) and MCB2028424 (Q.N.), and The Simons Foundation (594598 to Q.N.) of USA. T.L. is also partially supported by the Beijing Academy of Artificial Intelligence (BAAI). P.Z. also received the support from Study Abroad Program and Elite Program of Computational and Applied Mathematics for Ph.D. students of Peking University.

## Author contributions
Q.N., T.L., and P.Z. conceived the project; P.Z. and T.L. designed the algorithm and wrote the code; P.Z. and S.W. conducted the data analyses; P.Z. wrote the supplementary material; all the authors wrote and approved the manuscript. Q.N. and T.L. supervised the research.

## Competing interests
The authors declare no competing interests.
