## [Peer Review File · Nature Communications]

Dissecting transition cells from single-cell transcriptome data through multiscale stochastic dynamicsEditorial Note: This manuscript has been previously reviewed at another journal that is not operating a transparent peer review scheme. This document only contains reviewer comments and rebuttal letters for versions considered at *Nature Communications*.

REVIEWER COMMENTS

Reviewer #2 (Remarks to the Author):

The authors have satisfactorily addressed my comments from my previous reviews and I believe the clarifications and additions for scalability, this manuscript is suitable for publication in *Nature Communications*.

Reviewer #3 (Remarks to the Author):

1. SUMMARY

In their manuscript "Dissecting Transition Cells....", Zhou et al propose a new approach for analyzing the dynamics of single cell gene expression (transcriptome state) from static scRNAseq data. The authors present a method to identify "Transition Cells" – cell that are in the state of transitioning between high-dimensional attractor states. Their model is rooted in stochastic differential equations (overdamped Langevin equation, OLE) that describes the dynamics of cell state transition. They establish a connection between the SDE and the nearest neighbor graph of cells in scRNAseq (as shown by Lafon and Coifman [11]). They use the transition matrix estimated from the NN-graph of the single-cell data as a proxy for SDE dynamics.

The authors' main contribution in the method for identifying "transition cells" is the coarse-graining the dynamics from cell-cell (actually, transcriptional state - transcriptional state) dynamics to attractor-attractor (cluster-cluster) dynamics. In fact, the authors assume that the true global dynamics are governed by "coarse" attractor-attractor dynamics and local within-attractor dynamics and the observed finer-grained cell-cell transition process is a consequence of this system.

In this spirit, the author fit the coarse model to the observed cell-cell transitions (thereby determining attractor locations and attractor transition probabilities).

From this fitted model, various dynamics quantities of interest, MPFT, MPPT, and TCS are derived, on the basis of which the transition cells are determined. Then genes expressed during the transition (based on associated expression can be extracted.

In the remainder of the manuscript the authors demonstrate their method on various scRNAseq datasets and also compare the results with existing ad hoc algorithms, such as those that compute a pseudotime, etc. to identify transition cells.

Finally, I must also say: Despite the theory used, this work is not a rigorous evaluation of the question whether cell fate behavior revealed by single-cell complies with a fundamental theory. Yet it is more than the oft observed loosely theory-inspired approach. The theory plays a substantial role in the motivation of the method, but not as much as in the epistemological ideal of experimental verification of a theory.

=====

2. GENERAL REMARKS

I am most delighted to see an approach that analyzes scRNAseq data in terms of theoretical first principles of physics/dynamical systems. This is a refreshing departure from the current flood of ad-hoc heuristic methods that lack theoretical justification and are based on some hand-waving model, as are the vast majority of single-cell transcriptome analysis methods.

[Unfortunately, the authors do not make this clear upfront to differentiate themselves and only calls the latter "intuitive" approaches in the discussion.]

The method is sufficiently well described, but relies too much on details presented in the Methods section and even worse in the Supplement. The main text alone is not sufficient to understand the overall idea nor the complex implementation (epistemology of the overall framework of having a theory and linking scRNAseq data to the theoretical principles).

A strength is also that the authors made the code and datasets available online, which allowed me to check and analyze a few things myself.

The nice attempt to ground the work on a theoretical foundation makes this work stand out from among all the other computer science/statistics driven, ad hoc, heuristic and often rather pedestrian approaches. Nevertheless, this work by necessity (insufficient data, general lack of knowledge of system equations in biology) and thus, understandably, still had to employ a lot of ad hoc heuristics in the data preprocessing to connect the theory to the data.

Therefore, in evaluating the strength of this ms. one has to divide the assessment of quality of the innovation into two parts: (A) THE SCIENTIFIC THEORY: Is the grounding on first principles of non-equilibrium state transitions "correct" (the equation ...)? and (B) THE DATA ANALYTICS: Once we accepted the theoretical foundation, is the heuristics in connecting the theory to the data,, given all imperfection of experimental reality, appropriate? Moreover, in (C) I will comment on specific points in the application of the new method to published data sets .

=====

3. SPECIFIC CONCERNS

(A) SCIENTIFIC THEORY: <sorry, symbols got messed up in this textbox...>

A1. As said, it is truly refreshing to see a single-cell RNAseq analysis method that is motivated by scientific first principle: The stochastic non-linear dynamical and non-equilibrium systems framework. This affords this work a substantial strength and differentiation from all the other single-cell transcriptomics papers that (with a few notable exceptions) employ ad hoc computational/statistical heuristics and clever algorithmic hacks. The grand question here is: Is the assumption that allowed the use of the theory correct?

I have some doubts– but the authors' assumption is also a profound proposition that at least need careful introduction, justification, and if possible, even evaluation – none of which is provided. Specifically, the authors assume a gradient system, thermodynamic equilibrium in which detailed balance is satisfied:

They start, without explicit motivation, with the premise:

$$dX = -\text{grad}(U(x)) dt + \text{noise. (where noise is } \sqrt{2\epsilon} dW)$$

Accordingly, they assume $\delta(U)$ to be related to the transition rate via Kramer's theory; and assume that the probability distribution is related to U via Boltzmann-Gibbs.

This is already much assumed here. It is an approximation, albeit it may be a good one, but the author must be explicit about all this.

Thus, the authors could have started more generally with the dynamical systems description where the (overdamped) driving force is not a gradient of a potential. Then we have instead:

$$dX = F(x) dt + \text{noise.}$$

Then, one can assume that $F(x)$ has a component that is a gradient field of some potential, $V(x)$:

$$F(x) = -\text{grad}(V(x)) + f(x).$$

Note that $V(x)$ is not the same as $U(x)$ as used by the authors, but may approximate it (e.g. see PMID: 22933187)

If $[\text{grad}(V(x)), f(x)] = 0$, then and only then is $V(x)$ related to the transition rate for the transition $x_i \rightarrow x_j$ via " $P(x_i \rightarrow x_j) = \exp(-\Delta V/\epsilon)$ " – according to Freidlin-Wentzell.

Note that $V(x) \neq U(x)$, if $U(x)$ is, as the authors define, related to the STEADY-STATE probability (x) via the Boltzmann: $-\ln(U(x)) = P(X)$.

In other words, the authors assume detailed balance and thermodynamic equilibrium and steady-state distribution (of cells).

Living systems are open, far-from thermodynamic equilibrium systems that exhibit multiple non-

equilibrium stable steady-states (“attractors”). For the underlying molecular processes, i.e. at the “microscopic” level, detailed balance is violated.

Thus, in non-equilibrium systems the “barrier height” ΔU that needs to be overcome for attractor transition, is not the same as that determined by the probability density (Boltzmann), which is for steady-state and thermal noise, and is not for the Freidlin-Wentzell-quasi-potential for large perturbation-driven transitions not necessarily in steady-state. Thus, it seems that the authors also tacitly collapse the notion of non-equilibrium fluctuations, such as random bursts in gene expression (“gene expression noise”) with thermal noise.

Now to be scholarly generous one could argue in support of the authors’ assumption of a gradient system with detailed balance: For, one can find for some non-chaotic far-from equilibrium systems that $|\text{grad}(V(x))| \gg f(x)$. How generic this is in living systems, we do not know. Then one can assume $U(x) \approx V(x)$ which may apply to large regions in state space as has been shown in space region. (see: PMID: 32275714)

So here is an intriguing twist: It could well be that the “multiscale approach” takes care of the authors’ assumptions. Since quasi-stationary stable states in far-from equilibrium systems, while detailed balance is violated at the microscopic scale, it can be recovered at a larger (“mesoscopic”) scale. However, I doubt that this is the reason for the authors’ multi-scale approach, at least they do not elaborate. It is not at all clear how to frame single-cell state dynamics with fluctuating cell state (gene expression configuration) in terms of micro and mesoscale, so as to justify recovery of detailed balance.

In brief, the authors make rather big (related) assumptions without sufficient justification: (1) detailed balance satisfied; (2) potential differences (barrier heights) from Boltzmann STEADY-STATE probability distribution determines transition probability. BUT The reality is: The DEVELOPING cell population in many of the examples clearly are not in NOT in steady state. And even in steady-state, since we have thermodynamic non-equilibrium, microscopically, detailed balance is violated.

The authors’ method, clearly “works” as evidenced by their examples – however approximately (there is no benchmark – see below). Therefore, continuing with intellectual generosity here, which I think is appropriate in the service of scientific progress, we can accept the authors’ assumptions.

 But at the very least the authors must explicitly state the assumptions (declaring them as such) and succinctly and clearly also articulate upfront the possible justification for their assumptions, etc... This would much strengthen the ms.

A2. In the same vein, a manifestation of the authors’ subliminal assumption of classical thermodynamics, is their use of the term “METASTABLE state” to describe cell state attractors. This is confusing! In physics/chemistry Metastability implies that thermodynamic equilibrium is only prevented from being reached because of kinetic constraints. Entire living organism can be considered a metastable state (see theories of life as anti-catalysis etc). This reveals that the authors think in categories of classical thermodynamics. We have to be more precise here and operate with concepts independent of classical thermodynamics, but from DYNAMICAL SYSTEMS theory – that in a first approximation is independent of the former. Herein cell types have long been considered “attractors” (or “attractor states”). Rene Thom and Steve Small introduce this term in the late 60s – and we should stick to it and not use the term “metastable” to describe attractors in dynamical systems. Another reason is that it is confusing that the term “metastable” in medicine and biology has a different meaning: “Metastable” is often used by practitioners to indicate a shallow attractor, or even, an unstable steady-state – that is, a configuration that while stationary, is sensitive to perturbations.

(B) THE COMPUTATIONAL APPLICATION OF THE THEORY TO scRNA-SEQ DATA

Clearly at this second level of novelty of this ms, there are many gaps, but most are due to the intrinsic limitations of biological experimentation, thus generic to the field. Indeed in connecting theory with real-world data, even a theory-based paper must admit heuristic approaches, ad hoc assumptions... like the flurry of non-theory based computational approaches that we have seen in

the past years. Here one can have many options for cleaning the data and implementing particular algorithms, and the actual intrinsic merit is hard to establish – since the benchmark is often just: what works, works. The burden of demonstrating merit of a choice is therefore in the rigor in the latter step. Overall I found the authors' choices throughout reasonable, but there are a series of weaknesses that can be addressed.

B1. First and foremost, in the cell clustering, one must a priori decide on a number K of metastable states in the dataset. The authors use the "Eigen-Peak Index" as a criteria. (i) Is this a well-established quantity? I'm only familiar with the "eigengap" from spectral clustering. (ii) Looking at the eigen-peak index plots for the various datasets in the paper, it is apparent that the choice of clusters is far from clear/obvious. How robust are the results with respect to the number of clusters chosen, i.e. if one chooses smaller K , does the algorithm just yield more coarse grained dynamics or qualitatively different results?

B2. It is hard to tell if the results make sense in light of the data, since one can fit any model to any data. While the statistical metrics of goodness-of-fit themselves have issues, some qualitative, commonly accepted analysis would be appreciated: Plot the found clusters on the UMAP. Plot some of the inferred MS/IH/TD genes projected onto the standard UMAP layout. Allow the reader to inspect such displays for plausibility. It would strengthen confidence in the method.

B3. What is the purpose of the "dynamical manifold" (e.g. Fig2 b, Fig3c)? For visualization purposes I find it more confusing than helpful. These pseudo-3D plots, while attractive at first glance, don't convey much information. The transition path graphs (e.g. Fig. 5c) look more useful (keeping in mind that the whole thing is a forced 2D projection with several assumptions/simplifications as detailed in the Supplement)

B4. The authors compare their results to existing methods. However, it is very difficult to determine if one method is "better" than the other, without any ground truth and benchmark motivated by it. The highlighted differences might be circumstantial without greater meaning. As a side-note: Many of the analyzed datasets have a hierarchical structure of states (developmental trees), and the authors explicitly use tree structures for path-finding etc. So their method has an obvious advantage on these datasets (by encoding the correct inductive bias for these datasets). Hence a better performance (in terms of the expected biology, tree-like attractor structure) is a priori expected – and this has nothing to do with the validation of the underlying theory of transition states. This caveat should be addressed by reconfiguring the comparisons to make them "fair".

B5. An obvious competing method is PBA (Weinreb et al), which is based on similar theory (SDEs, and "fitting" the SDE to the cell-cell neighbourhood graph). It would be instructive to compare results obtained by both methods (PBA being the more "raw" method, while MuTrans applies all that coarse graining).

In all, as said above, the methods theoretical foundation per se are important enough that I consider MuTrans worthwhile to use because of the rationale and not because it performs better than other methods without a theoretical foundation. The comparison presented here to existing (rather ad hoc) methods is difficult, and at some point questionable.

C. SPECIFICS ON THE ANALYSIS OF THE VARIOUS DATA SETS

Yet, to more convincingly establish utility (not to say "correctness") the method for identifying transition cells the application to datasets could be refined/reviced, as detailed in the next sections. In general, here perhaps we have the situation of "less is more": I suggest to rather focus on a few "high-quality" dataset (many cells, clear-cut biology) and analyze them thoroughly instead of a superficial analysis of many datasets.

C1. The toy model. Here the authors simulate a system that exhibits two stable states (in a certain parameter regime) and infer the underlying structure with their algorithm. This is the only dataset where a ground truth is available, but unfortunately the authors failed to recognize the opportunity to compare their results against this ground truth. Several other things remain unclear:

- State transitions are induced by forcing an external parameter of the underlying SDE, i.e. the SDE is integrated (simulated) forward in time while slowly (?) changing the external parameter. This seems fundamentally different from a scenario where the parameter is fixed in the bistable regime, and the SDE solution jumps between the two stable states (triggered by noise). In particular this second scenario would allow for transitions in both directions (low->high, high->low), hence a more complicated setting for inference; the former scenario allows only one transition (easier for inference).
- How many data points (cells) are used here?
- How is the inference algorithm applied? For example, is the number of stable state $K=2$?
- Are the inferred quantities (transition rates etc) in agreement with the ground truth?

C2. EMT data. Here, the algorithm is applied to scRNAseq data of EMT. The author report three metastable states and analyze the genes involved in the transition process.

- A conventional display of the data (UMAP) would be helpful for this dataset.
- The number of clusters ($K=3$) is determined by the eigengap of the transition matrix. How robust is this? Standard algorithms (e.g. Louvain clustering) based on graph connectivity will yield far more clusters (~6 in case of the EMT data).
- What is this third metastable state (besides M and E)? The authors describe it as "low expression state". Has this been documented in the original publication of the data? Looking at the dataset briefly, to me it seems that these "cells" might either dead or lost most of their RNA content (less than 200 genes expressed). Moreover a flurry of recent publications, theoretical modeling (PMID: 26258068) and experimental (PMID: 26020648) have postulated/shown additional intermediate states in EMT.

C3. IPSC differentiation induction. Applied to the IPS dataset, MuTrans detects 9 metastable states and infers transition paths between these states. The points raised above for the EMT dataset apply here to:

- UMAP display of the data, colored by clusters identified to offer the reader a sense of the clusters
- Looking at Figure S7d, the number of clusters $K=9$ seems almost arbitrary. There's an "apparent peak" also at $k=4$, $k=7$ etc... Furthermore, while the authors choose $K=9$, Figure S7c show 8 states!
- about these gene groups (MS/IH/TD): I would expect the "metastable genes" to be "cluster markers", i.e. either exclusively expressed in that particular metastable state, or at least show a sharp expression change between that metastable state and the neighboring states. However, this does not seem to be the case. For example, looking at the M-state and its MS genes (Suppl Table S6), MSX2 is expressed all over the place (except day0 and day 1 cells). GATA5 is expressed in both M and En states, same as ISL1. On the other hand clear marker genes (MYL4) are not listed as MS
- Figure S9, center plot: The distinction between "Stable PS markers" and "Pre-En" ("Pre-M") markers seems arbitrary: For example, GATA6 is deemed a PS-marker (although it is expressed in Pre-En and Pre-M cells), while BAMB1 (which has pretty much the same expression profile across cells) is called a "Pre-M marker".
- The paper form which the data was used also has presented a way to characterize cells in transition ("pre-bifurcation") based on a particular quantity I. The authors could compare to see how that quantity relates to the Transition Cell Score (TCS) that they propose.

C4. Myelopoiesis/ Lymphopoiesis

The authors identify 10 metastable states. How robust is this, given the fact that the entire dataset only contains 375 cells! I find it hard to believe that this complicated hierarchy in Figure S10 can be reliably inferred from this small dataset.

- Similarly, for the lymphoid dataset, the choice of cluster number seems arbitrary; that choice is hard to motivate given the EigenPeak plot in Figure S14a. Any subsequent biological interpretation (page 13 main text of manuscript) is thus questionable, and should be worded accordingly.

C5. Human HSCs. The number of clusters seems to be chosen as $K=5$. This seems a reasonable choice compared to previous examples (which had far less cells than this dataset).

Some more notes on Comparison: It is hard to compare these various algorithms without some known ground truth. Results will always differ slightly, and it is unclear which method (if any) is

correct. Furthermore, slight differences in preprocessing affects results significantly. Just as one example: For the iPSC-dataset, normalizing cells by RNA content (a fairly standard step) resolves PAGAs issue of "short-circuiting" ectoderm to later stages.

Response to Reviewer #2 (Remarks to the Author)

The authors have satisfactorily addressed my comments from my previous reviews and I believe the clarifications and additions for scalability, this manuscript is suitable for publication in Nature Communications.

Authors' Response

We are glad to hear this, and we thank the reviewer for the insightful comments and helpful suggestions in previous rounds of revisions.

Response to Reviewer #3 (Remarks to the Author)

1. SUMMARY

In their manuscript "Dissecting Transition Cells...", Zhou et al propose a new approach for analyzing the dynamics of single cell gene expression (transcriptome state) from static scRNAseq data. The authors present a method to identify "Transition Cells" – cell that are in the state of transitioning between high-dimensional attractor states. Their model is rooted in stochastic differential equations (overdamped Langevin equation, OLE) that describes the dynamics of cell state transition. They establish a connection between the SDE and the nearest neighbor graph of cells in scRNAseq (as shown by Lafon and Coifman [11]). They use the transition matrix estimated from the NN-graph of the single-cell data as a proxy for SDE dynamics.

The authors' main contribution in the method for identifying "transition cells" is the coarse-graining the dynamics from cell-cell (actually, transcriptional state - transcriptional state) dynamics to attractor-attractor (cluster-cluster) dynamics. In fact, the authors assume that the true global dynamics are governed by "coarse" attractor-attractor dynamics and local within-attractor dynamics and the observed finer-grained cell-cell transition process is a consequence of this system. In this spirit, the author fit the coarse model to the observed cell-cell transitions (thereby determining attractor locations and attractor transition probabilities).

From this fitted model, various dynamics quantities of interest, MPFT, MPPT, and TCS are derived, on the basis of which the transition cells are determined. Then genes expressed during the transition (based on associated expression can be extracted.

In the remainder of the manuscript the authors demonstrate their method on various scRNAseq datasets and also compare the results with existing ad hoc algorithms, such as those that compute a pseudotime, etc. to identify transition cells.

Finally, I must also say: Despite the theory used, this work is not a rigorous evaluation of the question whether cell fate behavior revealed by single-cell complies with a fundamental theory. Yet it is more than the oft observed loosely theory-inspired approach. The theory plays a substantial role in the motivation of the method, but not as much as in the epistemological ideal of experimental verification of a theory.

Authors' Response

We thank the reviewer so much for the careful and thorough reading of our manuscript, and we appreciate the insightful comments and helpful suggestions. We appreciate the reviewer's comments that the purpose of this work is not to validate the dynamical system theory of cell-state transition from single-cell data, but instead propose a new data analysis strategy that is inspired and rooted from the theory.

By carefully addressing reviewer's comments and suggestions in later text, we believe that the manuscript has been improved significantly. Below is a summary of the major revisions we've made in this version:

1. Regarding the theoretical aspect, we focus on stating and explaining the underlying assumptions of MuTrans more explicitly and rigorously, especially in the main text.
2. Regarding the computational aspect, we clarify several confusions relevant to EPI, dynamical manifold and MS/IH gene issues, and we also adjust the method comparison sections by following the reviewer's suggestions.
3. Regarding the data analysis part, we follow all the specific instructions suggested. Especially we create two new simulation datasets, re-analyze the EMT dataset in greater details and clarify the previous confusions in IPS dataset analysis. We also update our codes to make MuTrans seamlessly compatible with mainstream scRNA-seq analytical tool Scanpy, making the combination with other analysis (UMAP dimension reduction, gene analysis) more automatic and convenient.

=====

2. GENERAL REMARKS

I am most delighted to see an approach that analyzes scRNAseq data in terms of theoretical first principles of physics/dynamical systems. This is a refreshing departure from the current flood of ad-hoc heuristic methods that lack theoretical justification and are based on some hand-waving model, as are the vast majority of single-cell transcriptome analysis methods.

[Unfortunately, the authors do not make this clear upfront to differentiate themselves and only calls the latter "intuitive" approaches in the discussion.]

Authors' Response

Thank you for the positive comments on the novelty of our methods. We apologize for not presenting the theoretical characteristics of MuTrans explicitly enough in previous version. Please see below on our actions to address this point.

The method is sufficiently well described, but relies too much on details presented in the Methods section and even worse in the Supplement. The main text alone is not sufficient to understand the overall idea nor the complex implementation (epistemology of the overall framework of having a theory and linking scRNAseq data to the theoretical principles).

Authors' Response

Thanks for the comments about method presentation. We agree that the more thorough explanation of the overall idea and implementation framework of MuTrans in main text will be helpful for the readers. Please see below on our actions to address this point.

A strength is also that the authors made the code and datasets available online, which allowed me to check and analyze a few things myself.

The nice attempt to ground the work on a theoretical foundation makes this work stand out from among all the other computer science/statistics driven, ad hoc, heuristic and often rather pedestrian approaches. Nevertheless, this work by necessity (insufficient data, general lack of knowledge of system equations in biology) and thus, understandably, still had to employ a lot of ad hoc heuristics in the data preprocessing to connect the theory to the data.

Authors' Response

Thank you again for the positive comments on the strengths of our method. We agree with the reviewer that when linking the abstract theory to actual datasets, certain ad-hoc, heuristic pre-processing procedures seem inevitable. We believe that by disclosing the data processing procedures, testing the robustness over some hyper-parameters, and explicitly discussing the limitations/possible improvements, the overall soundness of work can be well supported. This can also be useful to inspire future works along this direction. Please see below for more details on our revision to address this point.

Therefore, in evaluating the strength of this ms. one has to divide the assessment of quality of the innovation into two parts: (A) THE SCIENTIFIC THEORY: Is the grounding on first principles of non-equilibrium state transitions "correct" (the equation)? and (B) THE DATA ANALYTICS: Once we accepted the theoretical foundation, is the heuristics in connecting the theory to the data,, given all imperfection of experimental reality, appropriate? Moreover, in (C) I will comment on specific points in the application of the new method to published data sets.

Authors' Response

We agree with reviewer's evaluation criterion, and are grateful for these detailed, constructive suggestions. By addressing all the reviewer's comments as specified below, we believe that the manuscript has been significantly improved, including the aspect of data analysis.

=====

3. SPECIFIC CONCERNS

(A) SCIENTIFIC THEORY: <sorry, symbols got messed up in this textbox...>

A1. As said, it is truly refreshing to see a single-cell RNAseq analysis method that is motivated by scientific first principle: The stochastic non-linear dynamical and non-equilibrium systems framework. This affords this work a substantial strength and differentiation from all the other single-cell transcriptomics papers that (with a few notable exceptions) employ ad hoc computational/statistical heuristics and clever algorithmic hacks. The grand question here is: Is the assumption that allowed the use of

the theory correct?

I have some doubts— but the authors' assumption is also a profound proposition that at least need careful introduction, justification, and if possible, even evaluation – none of which is provided.

Authors' Response

We apologize for not presenting sufficient introduction or justification for the underlying theory in the original main text. By addressing the below points raised by reviewer, we believe that the assumption underlying MuTrans is better described with more details.

Specifically, the authors assume a gradient system, thermodynamic equilibrium in which detailed balance is satisfied:

They start, without explicit motivation, with the premise:

$$dX = -\text{grad}(U(x)) dt + \text{noise. (where noise is } \sqrt{2\epsilon} dW \text{)}$$

Accordingly, they assume $\Delta(U)$ to be related to the transition rate via Kramer's theory; and assume that the probability distribution is related to U via Boltzmann-Gibbs. This is already much assumed here. It is an approximation, albeit it may be a good one, but the author must be explicit about all this.

Authors' Response

Thanks for the suggestion about stating the underlying assumption of MuTrans explicitly.

Authors' Action

In addition to the previous description in Methods section, in the revised manuscript we include the explicit introduction about the underlying dynamics of MuTrans in the Introduction section of the main text (line 103).

Thus, the authors could have started more generally with the dynamical systems description where the (overdamped) driving force is not a gradient of a potential. Then we have instead:

$$dX = F(x) dt + \text{noise.}$$

Then, one can assume that $F(x)$ has a component that is a gradient field of some potential, $V(x)$:

$$F(x) = -\text{grad}(V(x)) + f(x).$$

Note that $V(x)$ is not the same as $U(x)$ as used by the authors, but may approximate it (e.g. see PMID: 22933187)

If $[\text{grad}(V(x)), f(x)] = 0$, then and only then is $V(x)$ related to the transition rate for the transition $x_i \rightarrow x_j$ via " $P(x_i \rightarrow x_j) = \exp\left[-\frac{\Delta V}{\epsilon}\right]$ " – according to Freidlin-Wentzell.

Note that $V(x) \neq U(x)$, if $U(x)$ is, as the authors define, related to the STEADY-STATE probability (x) via the Boltzmann: $-\ln(U(x)) = P(X)$.

In other words, the authors assume detailed balance and thermodynamic equilibrium and steady-state distribution (of cells).

Authors' Response

Thank you for the insightful comments about the detailed balance assumption of MuTrans. We agree that assumptions of detailed-balance and stationary distribution are the simplified approximation for the more general dynamical system governing the cell-fate transitions, where non-equilibrium effect can indeed be strong (such as oscillation dynamics or in fast-growing populations). In fact, similar assumptions (for example, stronger conditions about the concrete drift term) might be inevitable if only limited prior knowledge is given on the properties of the single-cell datasets. As shown by Klein et al. (PNAS 2018), usually the dynamical system is not uniquely defined if 1) the detailed-balance assumption is not enforced or 2) no additional knowledge is provided (such as RNA splicing dynamics, time points information or estimated cell growth rate) in the snapshot scRNA-seq dataset.

Authors' Action

In the revised Introduction, we start with the general form of dynamical system, and states that detailed balance + ergodicity (stationary distribution) are the two important underlying assumptions made by MuTrans, along with its rationale to guarantee the well-posedness of problem (line 113-123). We also add the reference [30-31] here to illustrate the rationales for the assumptions.

Living systems are open, far-from thermodynamic equilibrium systems that exhibit multiple non-equilibrium stable steady-states ("attractors"). For the underlying molecular processes, i.e. at the "microscopic" level, detailed balance is violated.

Thus, in non-equilibrium systems the "barrier height" ΔU that needs to be overcome for attractor transition, is not the same as that determined by the probability density (Boltzmann), which is for steady-state and thermal noise, and is not for the Freidlin-Wentzell-quasi-potential for large perturbation-driven transitions not necessarily in steady-state. Thus, it seems that the authors also tacitly collapse the notion of non-equilibrium fluctuations, such as random bursts in gene expression ("gene expression noise") with thermal noise.

Now to be scholarly generous one could argue in support of the authors' assumption of a gradient system with detailed balance: For, one can find for some non-chaotic far-from equilibrium systems that $|\text{grad}(V(x))| \gg f(x)$. How generic this is in living systems, we do not know. Then one can assume $U(x) \approx V(x)$ which may apply to large regions in state space as has been shown space region. (see: PMID: 32275714)

So here is an intriguing twist: It could well be that the "multiscale approach" takes care of the authors' assumptions. Since quasi-stationary stable states in far-from equilibrium systems, while detailed balance is violated at the microscopic scale, it can be recovered at a larger ("mesoscopic") scale. However, I doubt that this is the reason for the authors multi-scale approach, at least they do not elaborate. It is not at all clear how to frame single-cell state dynamics with fluctuating cell state (gene expression configuration) in terms of micro and mesoscale, so as to justify recovery of detailed balance.

Authors' Response

Thanks for these thought-provoking comments about detailed balance condition. We appreciate the reviewer's comments that under the condition when 1) gradient force in drift term is dominated 2) the dynamics is inspected from a larger time or space scale, detailed-balance assumption can be well accepted. In addition, the theoretical analysis in Supplementary Material (**SM section 1.2**) indicates that if the condition is satisfied, then the data-driven state-transition rates calculated by MuTrans is consistent with Kramer's law in the continuum limit. Of course, not all living system satisfies the condition, and MuTrans might confront with serious limitations in analyzing datasets including oscillation dynamics (e.g. cell cycle).

Authors' Action

In the revised main text Introduction, we explicitly add the reasoning for the assumptions (line 120-126). We comment on the scope of detailed balance condition (with the suggested paper as new reference [30]), and the limitations or possible extensions of MuTrans to analyze highly non-equilibrium systems in Discussion (line 477-486).

In brief, the authors make rather big (related) assumptions without sufficient justification: (1) detailed balance satisfied; (2) potential differences (barrier heights) from Boltzmann STEADY-STATE probability distribution determines transition probability. BUT The reality is: The DEVELOPING cell population in many of the examples clearly are not in NOT in steady state. And even in steady-state, since we have thermodynamic non-equilibrium, microscopically, detailed balance is violated.

The authors' method, clearly "works" as evidenced by their examples – however approximately (there is no benchmark – see below). Therefore, continuing with intellectual generosity here, which I think is appropriate in the service of scientific progress, we can accept the authors' assumptions.

 But at the very least the authors must explicitly state the assumptions (declaring them as such) and succinctly and clearly also articulate upfront the possible justification for their assumptions, etc... This would much strengthen the ms.

Authors' Response

Thank you again for all the helpful comments and suggestions above. This is a challenging task and requires significant new effort. In fact, we have on-going projects with goals to incorporate 1) non-equilibrium RNA splicing dynamics and 2) cell growth effect described by partial differential equations into our current framework. We hope to address these interesting issues in future publications.

Authors' Action

In addition to the revisions made above regarding the basic method assumptions, in Discussion we add the importance of including RNA velocity or/and temporal points in improving the application scope of MuTrans (line 482-486).

A2. In the same vein, a manifestation of the authors' subliminal assumption of classical thermodynamics, is their use of the term "METASTABLE state" to describe cell state attractors. This is confusing! In physics/chemistry Metastability implies that thermodynamic equilibrium is only prevented from being reached because of kinetic constraints. Entire living organism can be considered a metastable state (see theories of life as anti-catalysis etc). This reveals that the authors think in categories of classical thermodynamics. We have to be more precise here and operate with concepts independent of classical thermodynamics, but from DYNAMICAL SYSTEMS theory – that in a first approximation is independent of the former. Herein cell types have long been considered "attractors" (or "attractor states"). Rene Thom and Steve Small introduce this term in the late 60s – and we should stick to it and not use the term "metastable" to describe attractors in dynamical systems. Another reason is that it is confusing that the term "metastable" in medicine and biology has a different meaning: "Metastable" is often used by practitioners to indicate a shallow attractor, or even, an instable steady-state – that is, a configuration that while stationary, is sensitive to perturbations.

Authors' Response

We apologize for the confusion about the term "metastable states". We adopted this term because in the stochastic dynamical system community, the term "metastability" indicates the induced switching among different attractors (in the long-time scale and coarse space scale), a natural analogy for the cell state transitions. We agree with the reviewer that "metastable states" might have different meanings in physics/chemistry/biology, and we are grateful for the thorough introduction here. For the broader audience, we decide to follow the suggestion, changing the term "metastable states" to "attractors" throughout the revised manuscript.

Authors' Action:

1. In the revised manuscript, we now use the new term "attractors" instead of "metastable states" (and also attractor basins/potential wells, when we want to distinguish more "stable cells" near fixed point or transient cells near saddle points).
2. We also include the references about the notions of attractors in Introduction (ref [19-21]).

(B) THE COMPUTATIONAL APPLICATION OF THE THEORY TO scRNA-SEQ DATA

Clearly at this second level of novelty of this ms, there are many gaps, but most are due to the intrinsic limitations of biological experimentation, thus generic to the field. Indeed in connecting theory with real-world data, even a theory-based paper must admit heuristic approaches, ad hoc assumptions... like the flurry of non-theory based computational approaches that we have seen in the past years. Here one can have many options for cleaning the data and implementing particular algorithms, and the actual intrinsic merit is hard to establish – since the benchmark is often just: what works,

works. The burden of demonstrating merit of a choice is therefore in the rigor in the latter step. Overall I found the authors' choices throughout reasonable, but there are a series weaknesses that can be addressed.

Authors' Response

We appreciate the reviewer's evaluation logic and are grateful for the suggestions. By following these suggestions (described below), we believe that the demonstration of our algorithm application is now more rigorous and solid.

B1. First and foremost, in the cell clustering, one must a priori decide on a number K of metastable states in the dataset. The authors use the "Eigen-Peak Index" as a criteria. (i) Is this a well-established quantity? I'm only familiar with the "eigengap" from spectral clustering. (ii) Looking at the eigen-peak index plots for the various datasets in the paper, it is apparent that the choice of clusters is far from clear/obvious. How robust are the results with respect to the number of clusters chosen, i.e. if one chooses smaller K, does the algorithm just yield more coarse grained dynamics or qualitatively different results?

Authors' Response

We apologize for the confusions about "Eigen-Peak Index" (EPI). This index intends to describe the eigen-gap with the "peak" corresponding to the location of the gap.

Authors' Action

1. In the revised Method, we now add more detailed explanation about EPI (line 531-535).
2. In the newly generated simulation datasets, we now show that the eigen-peak is a good indicator on the number of attractors of potential field (SM Figure S4 (c)(f)).
3. In the real-world datasets, we now 1) test the robustness of transition cells/paths when the number of clusters is chosen by the different peaks in EPI (Figure S6-h and the associated Python notebook for this dataset) and 2) clarify that prior biological knowledge (e.g. expression of known markers or existing annotations in the dataset) can also serve as a reference to determine K, especially when the number of cells are limited (main text line 534, SM Section 2.7).

B2. It is hard to tell if the results make sense in light of the data, since one can fit any model to any data. While the statistical metrics of goodness-of-fit themselves have issues, some qualitative, commonly accepted analysis would be appreciated: Plot the found clusters on the UMAP. Plot some of the inferred MS/IH/TD genes projected unto the standard UMAP layout. Allow the reader to inspect such displays for plausibility. H is would strengthen confidence in the method.

Authors' Response

Thanks for the nice suggestions to project the MS/IH/TD on UMAP plots. In the revised main and supplementary figures, we now include the UMAP plot to display the data.

Authors' Action

Please see the specific actions for various datasets in C part below on this point. In our package, we also include the convenient interface with Scanpy to directly perform other downstream analysis (including dimension-reduction plots, gene marker detection and visualization) based on user's need. The usage examples are recorded in the Python notebooks (<https://github.com/cliffzhou92/MuTrans-release/blob/main/Example>).

B3. What is the purpose of the "dynamical manifold" (e.g. Fig2 b, Fig3c)? For visualization purposes I find it more confusing than helpful. These pseudo-3D plots, while attractive at first glance, don't convey much information. The transition path graphs (e.g. Fig. 5c) look more useful (keeping in mind that the thing is a forced 2D projection with several assumptions/simplifications as detailed in the Supplement)

Authors' Response

Sorry for the confusion about "dynamical manifold". It can be viewed as a reduced energy landscape (or quasi-potential) of the underlying dynamical system in *two dimensions*, which uses Gaussian mixtures to approximate the steady-state distribution. The z axis represents the "energy values", and x-y axis are the 2D projection of the dataset, consistent with Figure 5c. The manifold can intuitively illustrate the relative cell positions in attractor basins (near fixed point or saddle point), and the energy barrier between basins can also quantify the relative transition rate. These points are expanded in the new version. Although not perfect, we think the Gaussian mixture description shows the key characteristics of the distribution of cells in terms of the location of cell centers, accumulation pattern around a specific center, and the difficulty of transitions. It is very simple and presents useful information in a qualitative way.

Following the comments by Reviewer 2 in previous round of revision, we use the term "dynamical manifold" to distinguish from the term "low-dimensional geometric manifold" commonly applied in single-cell analysis.

We also agree with the Reviewer that the 2D projection version overlaid with transition paths can also be informative. Now we include them in the revised figures.

Authors' Action

1. In the revised Method, we expand the description on dynamical manifold and its relationship with the energy landscape, and add more references about the general concept of energy landscape/quasi-potential for the general readers (ref [31,44,45], line 711-713).
2. In main text Figures 2 and 3, we also include the 2D projection of dynamical manifold, overlaid with transition paths.

B4. The authors compare their results to existing methods. However, it is very difficult to determine if one method is "better" than the other, without any ground truth and benchmark motivated by it. The highlighted differences might be circumstantial without greater meaning. As a side-note: Many of the analyzed datasets have a hierarchical

structure of states (developmental trees), and the authors explicitly use tree structures for path-finding etc. So their method has an obvious advantage on these datasets (by encoding the correct inductive bias for these datasets). Hence a better performance (in terms of the expected biology, tree-like attractor structure) is a priori expected – and this has nothing to do with the validation of the underlying theory of transition states. This caveat should be addressed by reconfiguring the comparisons to make them “fair”.

Authors' Response

Thanks for the insightful comments about method comparison. We agree with the reviewer that MuTrans features for 1) multiscale dissection of the system underlying datasets and 2) distinguishing transient cells from stable ones in attractor basins, therefore the result is largely incomparable with majority of existing methods. Only certain downstream result of MuTrans, for example the transition path analysis, can be compared with other trajectory inference methods. Still, we agree that 1) fundamental difference in method assumption and 2) lack of exact ground truth in datasets make the strict and fair benchmarking hard to implement.

Hence in the revised manuscript, 1) we stress the unique feature about MuTrans and 2) in actual comparison of trajectory inference, we focus on the consistency rather than discrepancy between different methods. We use such consistency to support MuTrans, which is developed based on distinctive theories compared to other methods.

Authors' Action

1. In main text Result and Discussion about method comparison, we point out the special, model-inspired feature of MuTrans. Please see changes on lines 369-374, 406-408, 424-432, 434-440.
2. In main text Result (line 369-396) and Supplementary Material section 4, we show the consistency between MuTrans transition path analysis and some existing methods. We also show the unique feature of MuTrans to distinguish transition cells from stable ones.

B5. An obvious competing method is PBA (Weinreb et al), which is based on similar theory (SDEs, and "fitting" the SDE to the cell-cell neighbourhood graph). It would be instructive to compare results obtained by both methods (PBA being the more "raw" method, while MuTrans applies all that coarse graining).

Authors' Response

Thank you for the nice suggestion to PBA and we agree with reviewer's comments about their features. In revision, we now apply this method as a comparison (Figure S15,17), and discuss its relevance with MuTrans in Discussion (line 434-440). We also include analysis and discussions about another random walk based method Palantir (Setty et al.), and point out that the scopes of both methods are on the individual cell scale.

In all, as said above, the methods theoretical foundation per se are important enough that I consider MuTrans worthwhile to use because of the rationale and not because it

performs better than other methods without a theoretical foundation. The comparison presented here to existing (rather ad hoc) methods is difficult, and at some point questionable.

Authors' Response

Thanks again for the comments, and in the revised Introduction and Discussion, we now highlight the rationales and special features of MuTrans following reviewer's advice.

C. SPECIFICS ON THE ANALYSIS OF THE VARIOUS DATA SETS

Yet, to more convincingly establish utility (not to say "correctness") the method for identifying transition cells the application to datasets could be refined/reviced, as detailed in the next sections. In general, here perhaps we have the situation of "less is more": I suggest to rather focus on a few "high-quality" dataset (many cells, clear-cut biology) and analyze them thoroughly instead of a superficial analysis of many datasets.

Authors' Response

Thank you for these helpful and detailed suggestions for each dataset. In revision, we follow the suggestions to further explore simulation datasets as well EMT, IPS and human bone marrow datasets. Here is an outline of the revisions that we have made.

1. We created two new simulation datasets in considering back-forth transitions, and especially tested 1) the use of EPI to select number of attractors as the ground-truth is known and 2) the ability to recover "saddle points" (or cells in-transition) from the more stable cells around the fixed point of attractor basins.
2. We adopted the suggested pipeline to pre-process EMT dataset, and compared the intermediate cell states (ICS) to the previously annotated "hybrid EMT states", illustrating their roles in EMT transition paths.
3. We included more detailed analysis of IPS dataset, compared MuTrans analysis with a previously proposed bifurcation index, and clarified the notion of "meta-stable" (MS) genes.
4. We improved the main text by modifying some confusing descriptions as suggested.

C1. The toy model. Here the authors simulate a system that exhibits two stable states (in a certain parameter regime) and infer the underlying structure with their algorithm. This is the only dataset where a ground truth is available, but unfortunately the authors failed to recognize the opportunity to compare their results against this ground truth.

Several other things remain unclear:

- State transitions are induced by forcing an external parameter of the underlying SDE, i.e. the SDE is integrated (simulated) forward in time while slowly (?) changing the external parameter. This seems fundamentally different from a scenario where the parameter is fixed in the bistable regime, and the SDE solution jumps between the two

stable states (triggered by noise). In particular this second scenario would allow for transitions in both directions (low->high, high->low), hence a more complicated setting for inference; the former scenario allows only one transition (easier for inference).

- How many data points (cells) are used here?
- How is the inference algorithm applied? For example, is the number of stable state K fixed to $K=2$?
- Are the inferred quantities (transition rates etc) in agreement with the ground truth?

Authors' Response

Thanks for the nice suggestions on the toy model simulation. We agree that simulation models with back-forth transitions can better illustrate the rationale of our methods.

Authors' Action

1. Following the suggestions, we now include two more simulation datasets along with more detailed explanations. The two datasets are sampled from the over-damped Langevin dynamics in double-well or triple-well potential, respectively (allowing for the "jumps" between states), with each containing ~2000 data points (cells). We find that our method 1) correctly recovers the number of attractors using EPI, 2) identifies the consistent transition cells with the saddle points in the potential field, and 3) captures the transition rates. All the reproducible codes for the simulation datasets are recorded in the Matlab notebook files.
2. Accordingly, in main text Figure 2, we now supplement the original bifurcation example with the new analysis of triple-well potential simulation dataset, making all the simulation results as one stand-alone figure. Additional details on those simulations are added in the Supplementary Materials Figure S4.

C2. EMT data. Here, the algorithm is applied to scRNAseq data of EMT. The author report three metastable states and analyze the genes involved in the transition process.

Authors' Response

Thanks for the helpful suggestions about the EMT dataset. By addressing these issues (please see details below), we believe that the Mutrans analysis of this dataset is now significantly improved. All the revisions and reproducible codes are also recorded in the Python notebook file (<https://github.com/cliffzhou92/MuTrans-release/blob/main/Example/example-emt-raw.ipynb>).

- A conventional display of the data (UMAP) would be helpful for this dataset.

Authors' Action

In the revised main Figure 3 and SM Figure S5, we added the UMAP dimensional reduction plot of EMT dataset, overlaid with MuTrans results to highlight transition cells and gene expression patterns.

- The number of clusters (K=3) is determined by the eigengap of the transition matrix. How robust is this? Standard algorithms (e.g. Louvain clustering) based on graph connectivity will yield far more clusters (~6 in case of the EMT data).

Authors' Response

Thanks for the suggestion. After following the suggestion to remove low-expression cells (see our response to the next point), we find only one obvious peak observed at K =5, indicating that five attractors might be a desirable choice for this dataset, which is comparable with clustering based on community detection (5 clusters with default resolution = 1.0). We further annotate these attractors with gene expression patterns found in existing literatures (see our response to the next point).

Authors' Action

In the supplementary material Figure S5, we display the EPI plot of this dataset and the results by Leiden clustering, along with the marker gene analysis.

- What is this third metastable state (besides M and E)? The authors describe it as "low expression state". Has this been documented in the original publication of the data? Looking at the dataset briefly, to me it seems that these "cells" might either dead or lost most of their RNA content (less than 200 genes expressed). Moreover a flurry of recent publications, theoretical modeling (PMID: 26258068) and experimental (PMID: 26020648) have postulated/shown additional intermediate states in EMT.

Authors' Response

Thanks for the insightful comments and helpful references about "hybrid states" in EMT. In the revised manuscript, we now adopt the pre-processing techniques in the original publication to remove these "low expression" cells, limiting the minimum gene counts expressed and ERCC percentage within each cell. As shown in Figure S5, now five attractors are detected using such pre-processing, with one attractor expressing Epithelial markers (e.g. Epcam) – annotated as E state, two attractors expressing hybrid E/M markers (e.g. Zeb2, Prrx1) – annotated as ICS (Intermediate Cell State), and two attractors expressing mesenchymal markers (e.g. Mmp19 and Aspn) – annotated as M state. Interestingly, the transition path analysis by MuTrans reveals that the ICS plays the important mediation role in EMT.

Authors' Action

1. In the revised Result, we now add the more detailed explanation about the attractors detected in this dataset regarding the intermediate cell state (ICS), and

their relations with the “hybrid states” identified in previous literatures. We add the relevant references [35,36].

2. In the main text Figure 3, we now add information on the important roles of ICS dissected by the transition paths analysis of MuTrans.
3. In the Supplementary Material Figure S5, we plot the violin plot of known marker genes in different attractors to annotate the cell types.
4. In the Supplementary Material, we provide more details on the pre-processing steps, along with the Python notebook that allows reproduction of the results from the raw data (<https://github.com/cliffzhou92/MuTrans-release/blob/main/Example/example-emt-raw.ipynb>).

C3. IPSC differentiation induction. Applied to the IPS dataset, MuTrans detects 9 metastable states and infers transition paths between these states. The points raised above for the EMT dataset apply here to:

Authors' Response

Thanks for the helpful suggestions about the IPS dataset. All the revisions and reproducible codes are now recorded in the Python notebook file, as summarized point-by-point below.

(<https://github.com/cliffzhou92/MuTrans-release/blob/main/Example/example-emt-raw.ipynb>)

- UMAP display of the data, colored by clusters identified to offer the reader a sense of the clusters

Authors' Action

In the revised main Figure 4 and Figure S6, we now add the UMAP dimensional reduction plot of this dataset, overlaid with MuTrans results to highlight transition cells and relevant gene expression patterns. In the computing package, we now include the convenient interface with Scanpy that can directly perform other downstream analysis based on user's need.

- Looking at Figure S7d, the number of clusters $K=9$ seems almost arbitrary. There's an "apparent peak" also at $k=4$, $k=7$ etc... Furthermore, while the authors choose $K=9$, Figure S7c show 8 states!

Authors' Response

Thanks for the nice suggestion and we apologize for the confusions. Based on this, we add two clarifications for the EPI in the Methods. We point out that a) selecting different peaks correspond to specifying the resolution of scales that user wants to recover and b) determining the clustering resolution can depend on user's need and biological prior knowledge, for instance, the marker genes identified previously.

Specifically, we find that selecting $K=7$ will lose the resolution to uncover the “ectoderm” state and one of the initial states, and selecting $K=5$ will lose the distinguishment between two cell fates in the bifurcating (Pre-M and Pre-En). Now we add a study on the robustness and consistency of the identified transition cells and compute the entropy change before bifurcation and the transition paths toward M state.

However, determining the number of clusters is still a major challenge for any clustering methods. For instance, we find adjusting the “resolution” parameter in methods such as Louvian or Leiden graph partition can lead to different numbers of clusters.

Authors' Action

1. In the revised Method, we add more detailed explanation about the application of EPI. We point out that the peaks in EPI provide a reference for attractor numbers, and the actual choice can be combined with prior biological knowledge.
2. In the revised Supplementary Material Figure S6 (along with the Python notebook), we show concretely that for this dataset, changing the number of attractors (K) in MuTrans corresponds to different resolution on studying the process. The overall results on state-transition dynamics are found to be consistent regardless of the choice of the number of clusters.

- about these gene groups (MS/IH/TD): I would expect the "metastable genes" to be "cluster markers", i.e. either exclusively expressed in that particular metastable state, or at least show a sharp expression change between that metastable state and the neighboring states. However, this does not seem to be the case. For example, looking at the M-state and its MS genes (Suppl Table S6), *MSX2* is expressed all over the place (except day0 and day 1 cells). *GATA5* is expressed in both M and En states, same as *ISL1*. On the other hand clear marker genes (*MYL4*) are not listed as MS

Authors' Response

Sorry about the confusion of “meta-stable genes”. The “MS” genes are defined to reflect the relative trend of gene expression dynamics during the specific state-transition, rather than uniquely annotating the cluster as the “marker genes” in traditional sense. Such definition allows to distinguish stable attractor cells near the attractor points, at the bottom of basins and the transition cells between attractor basins, near saddles. The relative significance of each MS genes can also be quantified, which is indeed dependent on the threshold.

For this dataset, we agree that both M and pre-M state (even other states) express *MSX2*, while within M attractor, the expression of *MSX2* is lower in transition cells if the two attractors (Pre-M and M) are isolated from UMAP. In fact, *MSX2* is not a very significant MS genes according to our order (Table S4 in SM). For a more significant MS gene *TBX2* listed in the table, we observe the distinction between “stable” cells and transition cells in M attractor. Taken together, this indicates that the “MS” genes in M state can distinguish the more stable cells from the more transient cells, even they are in the same attractor basin, providing higher resolution for the transition process.

Authors' Action

1. In the revised Method, we now add a discussion on the difference between MS genes and traditional marker genes.
2. In Supplementary Material, we now add more description about how the genes are selected in Table S4 and their orders of significance.
3. In Supplementary Material Figure S7, we use UMAP plot and violin plot to show the expression pattern of MS and IH genes.

- Figure S9, center plot: The distinction between "Stable PS markers" and "Pre-En" ("Pre-M") markers seems arbitrary: For example, GATA6 is deemed a PS-marker (although it is expressed in Pre-En and Pre-M cells), while BAMB1 (which has pretty much the same expression profile across cells) is called a "Pre-M marker".

Authors' Response

Sorry about the confusion. In the heatmap, we plot all the cells *within* the PS state (instead of comparing cells in PS state to other states). The purpose is to distinguish the cells that are stable in the attractor basin and transiting toward Pre-En or Pre-M state, which are still in the PS attractor basin but closer to the saddle points, as sketched in the middle region of the landscape. All the cells displayed in revised Figure S8 middle heatmap correspond to the purple color cells in the schematic landscape.

Based on the UMAP plot of all the PS cells, GATA6 is up-regulated in transition cells compared to stable cells (therefore the low-expression of GATA6 can uniquely represent the "stable" PS cells). BAMB1 is up-regulated in the transition cells toward M cell-fate instead of En cell-fate (a more obvious example is MESP2, which is classified in the same group). In the heatmap, we now include another group genes, called non-bifur DE genes. These genes are exactly the "marker" genes in M or En cell-fate, although they cannot "predict" the bifurcation in PS cell attractor prior to the transition.

We agree with the reviewer that in this sense, the term "marker" might be confusing, and a name such as "bifur-predication genes" may be more appropriate.

Authors' Action

1. In the revised Figure S8, we now use boxes to show the cell correspondence from the gene expression heatmap to the schematic landscape.
2. To avoid confusion, we change the term "marker genes" to "bifur-prediction genes".
3. We add the UMAP plot of genes in PS cells to provide more illustration.
4. We now expand the description of this figure in the revised Supplementary Material (Section 3.3).

-The paper form which the data was used also has presented a way to characterize cells in transition ("pre-bifurcation") based on a particular quantity I. The authors could compare to see how that quantity relates to the Transition Cell Score (TCS) that they

propose.

Authors' Response

Thanks for this helpful and interesting idea to compare MuTrans with the bifurcation prediction index in the original literature. Since TCS is also defined for local transition process, for a better comparison globally we now add another index called "single-cell state transition entropy". The larger entropy values indicate stronger transient property of cells.

This newly defined entropy is consistent with the prediction index in the original paper to study the bifurcation. Inspecting the time-evolution of the entropy in this dataset, we observe that the entropy first increases as proceeding to the bifurcation states, and then decreases as the bifurcation process concludes.

Authors' Action

1. In the revised Method, we now add the definition of single-cell state transition entropy (line 595-605).
2. In the main text Figure 4 and Result, we add description on the consistency between MuTrans and the index in the original publication to study the bifurcation process (line 258-262).
3. In the Supplementary Material Figure S6, we add the above plot showing the trend of entropy change.

C4. Myelopoiesis/ Lymphopoiesis

The authors identify 10 metastable states. How robust is this, given the fact that the entire dataset only contains 375 cells! I find it hard to believe that this complicated hierarchy in Figure S10 can be reliably inferred from this small dataset.

- Similarly, for the lymphoid dataset, the choice of cluster number seems arbitrary; that choice is hard to motivate given the EigenPeak plot in Figure S14a. Any subsequent biological interpretation (page 13 main text of manuscript) is thus questionable, and should be worded accordingly.

Author's Response

We apologize for the inadequate description here. We agree that the number of cells are limited in these datasets compared with attractor numbers, while the biological annotations of the clusters are actually well supported or experimentally validated in the original publication. As a result, the cluster number selected in analysis depend on the prior knowledge in these two datasets.

Author's Action

1. In the revised Method, we now clarify that in addition to EPI, the biological prior is also an important reference to determine the number of attractors, especially when the number of cells are small compared to the desired number of cell states.

2. In the revised Result, we state that number of attractors are selected to recover the annotations in original publication (line 293-294). Indeed, both numbers correspond to one local peak of the EPI. Following reviewer's suggestion, we adjust our interpretations about new cell identities.
3. In Supplementary Material Figure S10, we plot the original labels to compare with our identified attractors.
4. In Supplementary, we plot the detailed transition paths analysis results (Figure S9). The results along with UMAP plots are also recorded in the new Python notebook. (<https://github.com/cliffzhou92/MuTrans-release/blob/main/Example/example-olsson.ipynb>)

C5. Human HSCs. The number of clusters seems to be chosen as $K=5$. This seems a reasonable choice compared to previous examples (which had far less cells than this dataset).

Author's Response

Thanks for the comment. Using this dataset, we also study the multi-scale reduction accuracy of MuTrans in analyzing the transition cells/paths (Figure 6, main text).

Some more notes on Comparison: It is hard to compare these various algorithms without some known ground truth. Results will always differ slightly, and it is unclear which method (if any) is correct. Furthermore, slight differences in preprocessing affects results significantly. Just as one example: For the IPSC-dataset, normalizing cells by RNA content (a fairly standard step) resolves PAGAs issue of "short-circuiting" ectoderm to later stages.

Author's Response

We agree that the pre-processing steps and parameter choice may largely affect the data analysis results. Now we clarify this point in the relevant parts in the revision with a focus on the special features of MuTrans.

REVIEWERS' COMMENTS

Reviewer #3 (Remarks to the Author):

The authors have adequately addressed my concerns both in the rebuttal letter as well as in the revisions of the manuscript.

Minor issues remain:

References (30) and (31) should be switched in order (concerning critical new paragraph around lines 121 - 123)

English; please distinguish between transiting (=passing through) Vs. transitioning (=making transition from A to B)

Line 243: poor English - do you mean "basins LOCATED BEFORE the bifurcation .."?

Line 257: English: What does "surpassing" mean here exactly?

Lines 481-484 ff: Too complex and speculative...OSCILLATION and GROWING CELLS are two fundamentally different types of departure from (quasi-) stationarity. The first is much harder to accommodate and comprehend intuitively in terms of potential functions because it deals with non-hyperbolic attractors... And I am not sure RNA velocity is the solution, since it is itself fraught with issues and makes many more assumptions. Also it is not clear what "cell-cell scale" means. Please simplify (just mention cell cycle and cell proliferation as other non-stationarity effects not considered, and reduce the speculation on possible solutions - it only distracts.. (but the point on the non-stationarity due to cell growth is good.)

I would have hoped that the authors drop the Myelopoiesis/Lymphopoiesis data set since due to the small number of cells, I am not confident about the data analysis there and here - but that is the author's choice. They also could more explicitly warn the reader.

Reviewer #3 (Remarks to the Author):

The authors have adequately addressed my concerns both in the rebuttal letter as well as in the revisions of the manuscript.

Author's Response: Thank you again for the careful reading and insightful comments on our manuscript. In this round of revision, we have followed all the suggestions to fix the minor issues and did a thorough proofreading of the whole draft.

The changes are highlighted with red colors in the manuscript, and the line numbers indicated below are accurate under the merged pdf files.

Minor issues remain:

References (30) and (31) should be switched in order (concerning critical new paragraph around lines 121 - 123)

Author's Response: Thank you for finding this mistake. We have switched their orders in the revised manuscript (line 124-126) and run the additional check of the references.

English; please distinguish between transiting (=passing through) Vs. transitioning (=making transition from A to B)

Author's Response: Thank you for the nice suggestion. We changed word transiting to transitioning at line 206.

Line 243: poor English - do you mean "basins LOCATED BEFORE the bifurcation .."?

Author's Response: Sorry for the confusions We changed the sentence as "Two attractor basins, locating before the bifurcation of..." (line 252)

Line 257: English: What does "surpassing" mean here exactly?

Author's Response: Sorry for the confusions. We modified the sentence as "...by passing through the pre-En attractor basins first." (line 266)

Lines 481-484 : Too complex and speculative...OSCILLATION and GROWING CELLS are two fundamentally different types of departure from (quasi-) stationarity. The first is much harder to accommodate and comprehend intuitively in terms of potential functions because it deals with non-hyperbolic attractors... And I am not sure RNA velocity is the solution, since it is itself

fraught with issues and makes many more assumptions. Also it is not clear what "cell-cell scale" means. Please simplify (just mention cell cycle and cell proliferation as other non-stationarity effects not considered, and reduce the speculation on possible solutions - it only distracts.. (but the point on the non-stationarity due to cell growth is good.)

Author's Response: Sorry for the confusions here and thanks for the nice suggestion. In this revision, we followed the suggestion to remove the previous point #1 and the excessive claims, and instead only mentioned that non-stationary effects due to cell cycle and cell growth were not considered by our current method (line 500-501).

I would have hoped that the authors drop the Myelopoiesis/Lymphopoiesis data set since due to the small number of cells, I am not confident about the data analysis there and here - but that is the author's choice. They also could more explicitly warn the reader.

Author's Response: Thank you for pointing out this and the helpful suggestions. The datasets were suggested by one of the reviewers in the previous rounds to study if the method can deal with datasets from different platforms and datasets with different properties. For myelopoiesis dataset, we mainly focused on the consistency with previous findings and other existing methods, and reported the observation that differences between MuTrans attractor basins and original labels could be potentially explained by transition cells.

We agree with the reviewer the data analysis is subject to the limited sample size. In the revision, we added the following sentences in Discussion to warn the readers:

“In addition, the number of cells in the datasets, in principle, needs to be sufficiently large in order to obtain high-resolution identification of transition cells. When the number of cells is relatively small, such as in the myelopoiesis dataset studied here, special care is needed to further confirm the analysis of transition cells.” (line 501-505)